# Pathogenic variants reveal candidate genes for prostate cancer germline testing for men of African ancestry

Prostate cancer (PCa) germline testing, while gaining momentum, is ancestry restrictive and African exclusive. Through whole genome sequencing for 217 African ancestral cases (186 southern African, 31 Pan representative), we identify 172 potentially pathogenic variants in 78 DNA damage repair or PCa related genes. Prevalence for reported (13/217, 5.99%) and cumulative predicted (24/217, 11.06%) variants of significance (11 genes) falls below that reported for non-Africans. Conversely, *BRCA1, HOXB13, CDK12, MLH1, MSH2*, and *BRIP1* remain unimpacted. Through pathogenic ranking based on variant frequency and functionality, clinical presentation and tumour-matched biallelic inactivation, top-ranked candidates include *PREX2, POLE, FAT1, BRCA2, POLQ, LRP1B* and *ATM*. Besides notable impact of DNA polymerases, including *POLG*, Fanconi anaemia genes include *FANCD2, FANCA, FANCG, ERCC4, FANCE* and *FANCI*, while DNA mismatch repair genes *MSH3* and *PMS1* outranked known namesakes *MSH6* and *PMS2*. This study provides insights into the spectrum of African-relevant potentially pathogenic PCa variants, highlighting much-needed gene candidates for ancestry-inclusive germline testing.

Germline testing (GT) for prostate cancer (PCa) is essential to optimise patients who benefit the most from precision medicine while predicting the risk of further malignancy for the patient and their relatives[1]. It encompasses testing for rare gene variants that are attributed to hereditary cancers, such as those involved in DNA repair[2]. With increased therapeutic implications[1,3], GT is moving beyond PCa risk assessment to include management of patients and screening of healthy men, as advocated by the National Comprehensive Cancer Network (NCCN) guidelines and other health organisations[2,4,5].

Besides a family history of PCa and younger age, African ancestry is a well-established risk factor for incidence, advanced disease and mortality[6,7]. However, guidelines for GT have almost exclusively been developed using non-African studies[2,8,9]. Recently, we showed that current GT panels are less optimal for rare pathogenic variants in South African patients of African ancestry[10], with prevalence only half of that reported for non-African populations (5.6% vs 11.8–17.2%)[11,12]. Concurring with previous, yet limited, African American and West African studies[13,14], we hypothesise that pathogenic variants mediating

the high-mortality pattern of PCa among African ancestral men are largely unknown. Notably, the lack of African-relevant data led the 2019 Philadelphia PCa Consensus Conference to exclude men of African ancestry from the current PCa GT criteria[8].

Representing globally the greatest PCa mortality rates[15] and home to genetically the most diverse populations[16], we initiated the Southern African Prostate Cancer Study (SAPCS), the founding study for the Health Equity Research and Outcomes Improvement Consortium Prostate Cancer Precision Health Africa1K (HEROIC PCaPH Africa1K)[17]. The overall aim - to generate African-relevant whole genome sequencing (WGS) data for the purpose of addressing PCa health disparities. Merging both published[18] and unpublished (this study) SAPCS data with Pan Prostate Cancer Group (PPCG) derived African ancestral WGS data[19] for a total of 217 cases, we use this unique data source to perform untargeted gene-wide interrogation for yet unknown potentially pathogenic variants. Here, we provide insights into potential gene candidates to establish PCa GT criteria for men of African ancestry.

✉ e-mail: vanessa.hayes@sydney.edu.au

## Results

### Genomic resources and patient characteristics

PCa cases recruited as part of the SAPCS or PPCG (Methods) from which blood-derived WGS germline data had been generated were sourced (Table S1). SAPCS data included 116 published[18], and 70 additional cases, the latter generated to an average of 43.3X coverage (range 36.4 to 69.1X), for a total of 186 South Africans of African ancestry. PPCG data (n = 990) was sourced from five countries, including Canada, Germany, United Kingdom, Australia (Melbourne and Sydney[18]), and France or French Caribbean, of which 31 cases are African ancestral[19]. WGS African-representative younger aged (<50 years) no cancer control data included 49 population-matched South Africans (southern African controls, SAC) and 40 Kenyans representing both east Bantu and Nilotic ethno-linguistic diversity (east African controls, EAC). Medical Genome Reference Bank (MGRB) WGS control data was sourced from 3,209 largely European ancestral Australians (1332 male, 1877 female) ≥75 years at time of recruitment and with no known cancer, hypertension or dementia[20]. Irrespective of data source, single-nucleotide variants (SNVs) and small insertions and deletions (indels; <50 base pairs) were called using GATK best practices.

Using 64,654 ancestry-informative SNVs, population substructure analysis was performed (Methods), confirming African ancestries for all 217 cases (Fig. 1A). At optimal k = 3 population inference (Supplementary Data 1), non-African fractions >10% are notably scarce for SAPCS (2.7%, 5/186; per patient range 12% to 64%) compared to PPCG patients (64.5%, 20/31, range 10.4% to 69.1%). Further, k = 4 defined SAPCS patients as southern African, with 68.8% (128/186) including southern African Khoe-San heritage (range 2% to 51.3%) (Fig. 1B). Ancestries within the PPCG are primarily west African derived (range 23% to 99.9%), with overall larger non-African fractions, as expected for Caribbean and African American patients, apart from a single PPCG patient with 52.9% southern African ancestry. SAPCS patients presented on average 2 years later (mean 66.7 years; range 43-99) compared with PPCG cases (mean 64.8 years; range 45-77) and with significantly advanced International Society of Urological Pathology Grade Group (ISUP) ≥4 (53.2% vs 19.4%, Chi-squared p-value < 0.0001) disease (Table S2). As previously reported[21], SAPCS men present with elevated Prostate-Specific Antigen (PSA) levels (mean 233.6 ng/mL; range 1 to 4,841) at almost 4-fold greater than PPCG Africans (mean 60.8 ng/mL; range 5 to 1150).

### Potentially pathogenic variants in African ancestral PCa patients

Nearly 59 million SNVs and 10 million indels from 217 African patients were interrogated for known potentially pathogenic variants (PPVs; Fig. 2 Step 1). Using the non-African biased ClinVar database, which includes the American College of Medical Genetics and Genomics and the Association for Molecular Pathology (ACMG-AMP) guidelines[22], pathogenic or likely pathogenic variants were identified and screened for all populations and African restricted minor allele frequency (MAF) using gnomAD v.4.0[23]. Consequently, 252 low-frequency inclusive PPVs were identified in 223 genes (195 SNVs, Supplementary Data 2 and 57 indels, Supplementary Data 3; 90 missense, 86 stop gain or loss, 22 splice variants, 51 frameshifts, 5 non-coding), of which 33 PPVs are absent from current databases (defined as unknown). Focusing on rare variants (MAF < 1%) resulted in 241 PPVs in 214 genes, with further Gene Set Enrichment Analysis (GSEA)[24] focused on genes associated with DNA damage repair (DDR) or PCa germline gene candidates, leaving 45 rare PPVs (11.11% unknown) in 34 genes (Table 1). Conversely, a single PPV in the DDR gene POLG p.Phe749Ser, while rare in population-wide global data (MAF = 0.0002) and absent in our African ancestral PPCG patients, presented at low frequencies in our SAPCS cases (MAF = 0.0134) and as such is classified here as a population-specific low-frequency (PSLF) PPV.

### Cross-ancestral correlations for African-relevant PPV-derived gene candidates

Focusing on the 223 genes harbouring low-frequency inclusive PPVs from African patients, we further interrogated for PPVs in 959 non-African PPCG cases and 3,209 MGRB healthy-aged controls (Fig. 2 Step 2). Here we identified 293 rare PPVs impacting 53.8% (120/223) of our African-derived gene candidates in 37.6% (361/959) non-African patients (Supplementary Data 4). Known PCa GT genes include BRCA2 (14 unique PPVs), ATM (7), CHEK2 (5), TP53 (3), RAD50 (2) and RAD54L (1). We also found PPVs in multiple African-relevant gene candidates including RECQL4 (5 unique PPVs), JAK2 (4) INO80 (3), EGFR (2), and ASPM (2), while APTX, LRP1B, FANCG, FANCD2, ERBB3, BUB1B, POLE, BLM, RAD54L, JAK3 and U2AF1 each presented with a single unique PPV each. Notable African-relevant GT candidate genes that lacked variance included TRRAP, CHD1L, ERBB4, MSH3, ROS1, PREX2, MYC, RET, CHD4, NF1, DONSON and STAG2. Additionally, 13 rare PPVs were shared between the ancestries (Table S3), including CHEK2 p.Arg283X and RAD50 p.Glu723fs, both previously known in PCa.

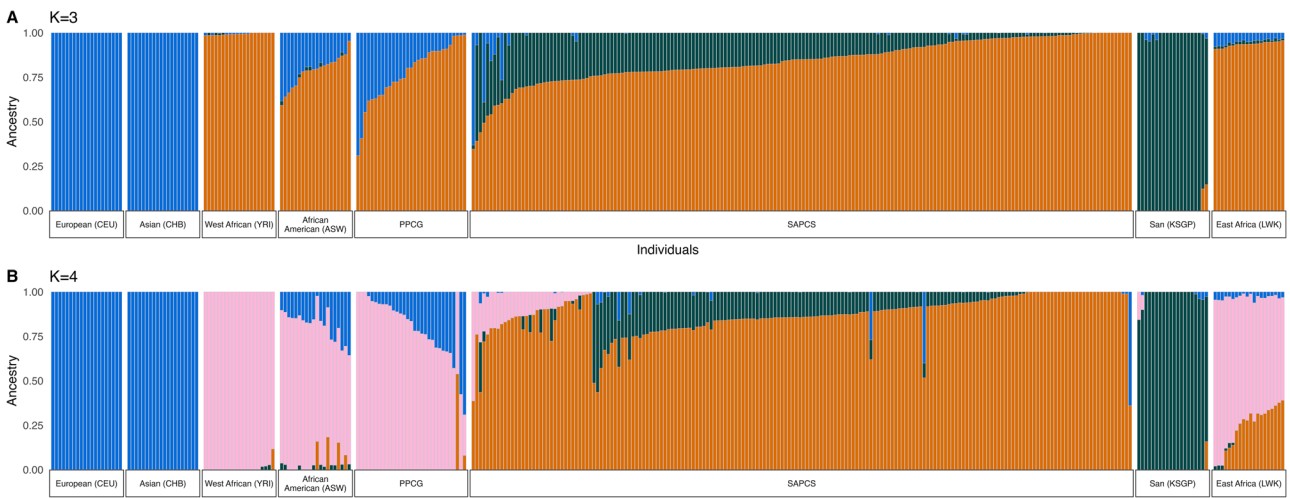

**Fig. 1 | Population genetic ancestral substructure for 217 African prostate cancer (PCa) cases.** Admixture plot for the study cohort including 186 South African (SAPCS) and 31 PPCG African ancestral patients using k-means clustering for k = 3 (**A**, cross-validation error = 0.252, Supplementary Data 1) and k = 4 (**B**, cross-validation error = 0.255, Supplementary Data 1). Population fractions have been determined against reference controls defined as; European (CEU), n = 20), Asian (CHB, n = 20), west African or Yoruba (YRI, n = 20), African American (ASW, n = 20), San (KSGP, n = 20) and east African or Luhya (LWK, n = 20).

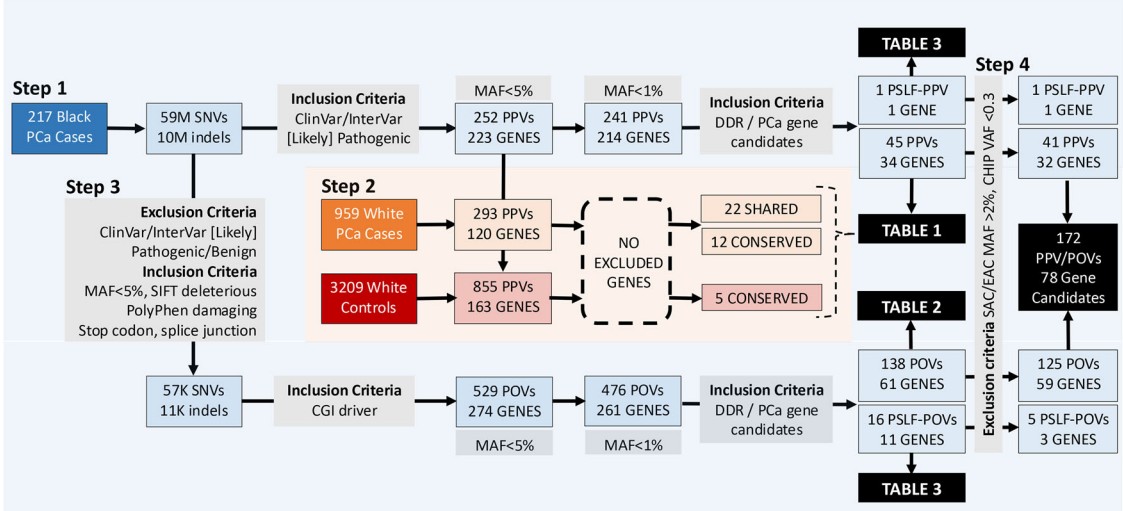

**Fig. 2 | Study workflow for the identification of African-relevant prostate cancer (PCa) Potentially Pathogenic Variants (PPVs) and Potentially Oncogenic Variants (POVs), including population-specific low-frequency (PSLF) PPVs/POVs and candidate genes. Step 1.** From genome-wide small variants (SNVs, single nucleotide variants; indels, insertions or deletions <50 bases) derived from 217 African PCa cases (blue) 45 rare DNA Damage Repair (DDR) or PCa related PPVs in 34 genes (Table 1) and a single PSLF-PPV (Table 3) were identified. **Step 2.** Rare and low-frequency PPV candidate genes (n = 223) were further filtered for non-African representative PPVs using European-biased PCa (PPCG, orange) and healthy (MGRB, red) datasets, provided multi-ethnic validation for 22 gene candidates,

genetic conservation for five genes and no further PPV candidate exclusion. **Step 3.** Prioritizing African-derived variants of unknown significance (VUS) for classification as POVs, as per exclusion and inclusion criteria (grey), yielded 138 rare DDR/PCa related POVs in 61 genes (Table 2) and 16 PSLF-POVs in 11 genes (10 overlapping with POV candidates, Table 3). Minor allele frequency (MAF) filtering (steps 1 and 3) was based on all population and African restricted gnomAD v4.0 data. **Step 4.** All class potential pathogenic variants were further filtered using population control MAFs >2% (SAC, southern African controls; EAC, east African controls) and variant allele frequency (VAF) < 30% for a total of 172 variants of pathogenic potential across 78 candidate genes.

For the healthy European ancestral population, we identified 855 rare PPVs impacting 74% (163/223) of gene candidates in 63.4% (2,004/3,209) of MGRB participants (Supplementary Data 5). The most abundantly impacted genes, although rare, include known PCa GT panel genes *ATM* (12 unique PPVs) and *CHEK2* (7), while African-relevant candidates included *EGFR* (13), *CHD4* (12), *ERBB4* and *RECQL4* (7 each). Of the 19 PPVs shared with our African PCa patients (Table S4), two impacted African-relevant candidates *RET* p.Val804Met and the *STAG2* splice variant rs1603095192G>T in a single individual each (MAF = 0.00016) and as such were not removed as candidates. In contrast, African-relevant PCa GT candidate genes *RAD54L, ROS1, LRP1B, JAK3* and *U2AF1* were highly conserved (lacked notable variance). Taken together, no genes were excluded based on the European population data.

### Characterising variants of unknown significance as potentially oncogenic

Low-frequency inclusive African variants of unknown significance (VUS), that are not in ClinVar and/or defined using ACMG-AMP criteria as pathogenic/likely pathogenic or benign/likely benign were further interrogated for oncogenic potential (Fig. 2 Step 3). After exclusion for common variants (MAF > 5%) found in all population and African restricted gnomAD data, VUS were maintained based on their functional potential defined as deleterious in SIFT[25], and/or damaging in PolyPhen-2[26], or disrupting a stop codon or splice junction, with additional oncogenic potential established using the Cancer Genome Interpreter (CGI)[27] providing the definition in this study as a potentially oncogenic variant (POVs). Identifying 529 POVs in 274 genes, after exclusion for common/low-frequency POVs (MAF > 1%) left 476 rare POVs in 261 genes (Supplementary Data 6). Focusing on DDR or PCa-associated genes, 138 rare POVs (15 unknown) remained in 61 gene candidates, including seven in known PCa GT-panel genes with an additional nine previously identified as PPV-derived candidates (Table 2), leaving 45 potential POV-derived candidate genes

(Supplementary Data 7), and 16 PSLF-POVs in 12 (11 overlapping with rare POV-derived) candidate genes (Table 3).

### Population-matched control and CHIP-associated filtering

As southern Africans are poorly represented in population databases such as gnomAD[28], we further sought to determine MAFs in healthy population-matched southern and east African controls (Fig. 2 Step 4). While three PPVs, *RAD50* p.Glu723fs (known to PCa), *TRRAP* p.Ala505fs and the inframe deletion identified in *RECQL4*, presented in a single East African (EAC MAF = 0.0116279), the latter including a single Southern African (SAC MAF = 0.0102041), none were excluded from further analyses (Table 1). Absent or negligible in all population or African restricted gnomAD data, 13 POVs were found to be rare in either SACs or EACs (single subject each) and as such were not excluded (Table 2 and Supplementary Data 7). Although *JAK2* p.Arg922Trp and *NDRG1* p.Ala84Ser were found in two SACs each (MAF=0.0204082), due to their absence from our EACs, we elected to cautiously maintain these POVs in downstream analyses, setting our MAF threshold for exclusion at >2%. As such, three POVs *ERCC6* p.Thr699Met (EAC MAF = 0.0465116), p.Ala906Gly (EAC MAF = 0.0348837) and *ERCC4* p.Ala860Asp (SAC/EAC combined MAF = 0.031915) were removed, leaving 135 POVs in 61 genes for further consideration.

Rare globally (through all population analyses) yet presenting at low frequencies within our African ancestral cases, the single PSLF-PPV and 13 of the 16 PSLF-POVs (81.25%) were restricted to our SAPCS cohort (Table 3). Notably, PSLF-POVs impacting known PCa GT panel genes *ATM* (p.Asp44Gly) and *PMS2* (p.Leu729X) presented in both our SACs and EACs (MAFs range 0.0204 to 0.0306) as did nine of the remaining PSLF-POVs and as such were removed from further analyses (Supplementary Data 8). Absent from SAC and EAC cohorts, besides the PSLF-PPV impacting the DDR gene *POLG* (p.Phe749Ser), the five remaining PSLF-POVs impacting the DDR-relevant oncogene *PREX2*[29] (p.Lys787Glu, p.Arg1230Trp, and rs150773140 slice donor) and DDR

**Table 1 | Rare Potentially Pathogenic Variants (PPVs, *n* = 45) identified in 217 African ancestral prostate cancer (PCa) patients impacting 34 DNA damage repair (DDR) or PCa related genes and as such further classified as known or candidate germline testing (GT) genes**

| Gene | Chr Position: nt Change | rsID[a] | AA Change | VAF[b] | SAPCS (n) | PPCG (n) | AFR Fraction[c] | MAF SAC (n = 49) | MAF EAC (n = 40) |
|---|---|---|---|---|---|---|---|---|---|
| BRCA2 [d] | chr13:32319100:T/C | rs80359182 | W31R | 0.439024 | 1 | 0 | >99% | 0 | 0 |
| | chr13:32337185:A/T | rs80358533 | K944X | 0.642857 | 1 | 0 | >99% | 0 | 0 |
| | chr13:32340123:ACATT/A | rs80359535 | I1924fs | 0.627907 | 1 | 0 | >99% | 0 | 0 |
| | chr13:32340123:CTG/C | rs80359478 | V1681fs | 0.4 | 0 | 1 | 73% | 0 | 0 |
| | chr13:32363421:T/G | rs80359070 | L2740X | 0.27907 | 1 | 0 | >99% | 0 | 0 |
| | chr13:32371013:AAAGG/A | rs397507406 | Q2850fs | ND | 0 | 1 | 89% | 0 | 0 |
| ATM [d] | chr11:108345818:C/T | rs587779872 | R2832C | ND | 0 | 1 | 87% | 0 | 0 |
| | chr11:108365476:C/T | rs121434219 | R3047X | All(2) > 0.4722 | 2 | 0 | >99% | 0 | 0 |
| RAD54L [d] | chr1:46250063:C/T | rs530382665 | R52W | ND | 0 | 1 | 57% | 0 | 0 |
| | chr1:46260594:C/T | rs149141765 | R64W | 0.461538 | 1 | 0 | >99% | 0 | 0 |
| RAD50 [d] | chr5:132595759:T/TA | rs397507178 | E723fs[e] | 0.526316 | 1 | 0 | >99% | 0 | 0.0125 |
| TP53 [d] | chr17:7673776:G/A | rs28934574 | R123W | 0.209302 | 1 | 0 | >99% | 0 | 0 |
| CHEK2 [d] | chr22:28734439:G/A | rs587781269 | R283X[e] | 0.478261 | 1 | 0 | >99% | 0 | 0 |
| NBN [d] | chr8:89978304:CA/C | rs1586101561 | C167fs | ND | 0 | 1 | 98% | 0 | 0 |
| RECQL4 | chr8:144513050:TG/T | rs1024114400 | P851fs | ND | 0 | 1 | 67% | 0 | 0 |
| | chr8:144513139:C/T | rs398124117 | Splice variant | 0.555556 | 1 | 0 | >99% | 0 | 0 |
| | chr8:144513260:ACGCCCGGCC/A | rs766312203 | RAGR804-807R | All(2) > 0.3541 | 2 | 0 | >99% | 0.0102041 | 0.0125 |
| FANCD2 | chr3:10047980:C/T | rs755992976 | Q448X | 0.530612 | 1 | 0 | >99% | 0 | 0 |
| | chr3:10085886:CAG/C | rs770686014 | Q1100fs | 0.49 | 0 | 1 | 88% | 0 | 0 |
| TRRAP | chr7:98910216:C/CA | rs1797006154 | A505fs | 0.137931 | 1 | 0 | >99% | 0 | 0.0125 |
| | chr7:98910234:CG/C | rs2116406452 | P512fs | 0.0909091 | 1 | 0 | >99% | 0 | 0 |
| CHD1L | chr1:147287651:G/A | unknown | W746X | 0.47 | 1 | 0 | >99% | 0 | 0 |
| ASPM | chr1:197121974:G/A | rs140602858 | R1271X | 0.648148 | 1 | 0 | >99% | 0 | 0 |
| LRP1B | chr2:140850147:G/A | rs1692413234 | R1632X | 0.517241 | 1 | 0 | >99% | 0 | 0 |
| ERBB4 | chr2:211383652:G/A | rs751834116 | P1297L | ND | 0 | 1 | 68% | 0 | 0 |
| MSH3 | chr5:80741503:TAATT/T | rs2112866803 | I537fs | 0.558824 | 1 | 0 | >99% | 0 | 0 |
| ROS1 | chr6:117310147:A/T | rs2128548191 | L2123H | 0.529412 | 1 | 0 | >99% | 0 | 0 |
| EGFR | chr7:55191740:C/T | rs371228501 | R831C | 0.56 | 1 | 0 | >99% | 0 | 0 |
| PREX2 | chr8:68022072:A/C | unknown | N125H | 0.444444 | 1 | 0 | >99% | 0 | 0 |
| MYC | chr8:127740711:C/T | rs2130105792 | T373I | 0.5 | 1 | 0 | >99% | 0 | 0 |
| JAK2 | chr9:5072609:C/A | rs149705816 | H587N | 0.465116 | 1 | 0 | >99% | 0 | 0 |
| APTX | chr9:33001566:C/A | rs146487634 | Splice variant | All(4) > 0.4 | 3 | 1 | >99% (3),92% | 0 | 0 |
| FANCG | chr9:35077266:TGGCGGTA/T | rs587776640 | YRQ213-215fs | All(2) > 0.42 | 2 | 0 | Both>99% | 0 | 0 |
| RET | chr10:43119548:G/A | rs79658334 | V804M[f] | 0.545455 | 1 | 0 | >99% | 0 | 0 |
| CHD4 | chr12:6601528:G/A | unknown | P180L | 0.48 | 1 | 0 | >99% | 0 | 0 |
| ERBB3 | chr12:56097837:G/A | rs771520731 | R838Q | 0.434783 | 1 | 0 | >99% | 0 | 0 |
| POLE | chr12:132649751:C/A | rs779261309 | E1241X | 0.613636 | 1 | 0 | >99% | 0 | 0 |
| BUB1B | chr15:40217665:T/C | unknown | Q964X | 0.45 | 1 | 0 | >99% | 0 | 0 |
| INO80 | chr15:40987186:C/T | rs199722402 | R1246Q | 0.48 | 0 | 1 | 90% | 0 | 0 |
| BLM | chr15:90749728:T/TG | unknown | W154Wfs | 0.44 | 1 | 0 | >99% | 0 | 0 |
| NF1 | chr17:31260369:GT/G | rs1555618803 | F1478X | 0.6 | 1 | 0 | >99% | 0 | 0 |
| JAK3 | chr19:17836001:G/A | rs149316157[e] | R613X | 0.47 | 1 | 0 | >99% | 0 | 0 |
| DONSON | chr21:33586090:A/G | rs1010722195 | F165S | 0.52 | 0 | 1 | >99% | 0 | 0 |
| U2AF1 | chr21:43094667:T/G | rs371246226 | Q157P[e] | 0.37 | 0 | 1 | 93% | 0 | 0 |
| STAG2 | chrX:124066174:G/T | rs1603095192 | Splice variant[f] | 1 | 1 | 0 | >99% | 0 | 0 |

*AA* amino acid, *AFR* African, *chr* chromosome, *EAC* east African Controls, *MAF* minor allele frequency, *ND* not determined, *nt* nucleotide, *PPCG* Pan Prostate Cancer Group, *SAC* southern African Controls, *SAPCS* Southern African Prostate Cancer Study, *VAF* variant allele frequency.

[a]Unknown rs-numbers are absent from variant databases.
[b]PPV exclusion based on CHIP-likelihood (VAF < 0.3).
[c]African ancestral genetic fraction presented as a percentage (southern, western and/or San) and *k* = 4 ADMIXTURE plot (Fig. 1).
[d]Known candidate germline testing gene.
[e]Present in a single European PPCG patient of 959.
[f]Present in a single European MGRB healthy control of 3209.

**Table 2 | Rare Potentially Oncogenic Variants (POVs) identified in 217 African ancestral prostate cancer (PCa) patients from the SAPCS (n = 186) and PPCG (n = 31) study cohorts and impacting 16 known and/or Potentially Pathogenic Variants (PPVs) recognised in DNA Damage Repair (DDR) germline testing genes**

| Gene | Chr position: nt change | rsID[a] | AA Change | VAF[b] | SAPCS (n) | PPCG (n) | AFR Fraction[c] | MAF SAC (n = 49)[d] | MAF EAC (n = 40)[d] |
|---|---|---|---|---|---|---|---|---|---|
| BRCA2 | chr13:32341158:G/A | rs80358906 | R2268K | 0.59 | 0 | 1 | 66% | 0 | 0 |
| | chr13:32357808:A/T | rs80358995 | F2562I | ND | 0 | 1 | >99% | 0 | 0 |
| | chr13:32370460:A/G | unknown | D2797G | 0.52 | 0 | 1 | 93% | 0 | 0 |
| PMS2 | chr7:5982890:G/A | rs370196722 | T703M | ND | 0 | 1 | 57% | 0 | 0 |
| | chr7:6003981:C/T | rs730881919 | E81K | 0.576923 | 1 | 0 | >99% | 0 | 0 |
| FANCA | chr16:89769976:C/T | rs771698195 | V879M | ND | 0 | 1 | 57% | 0 | 0 |
| | chr16:89783063:G/A | rs200291237 | R504G | 0.560976 | 1 | 0 | >99% | 0 | 0 |
| BARD1 | chr2:214767537:C/T | rs864622240 | G486R | 0.404255 | 1 | 0 | >99% | 0 | 0 |
| | chr2:214809476:C/G | rs1224914625 | G32R | 0.512195 | 1 | 0 | >99% | 0 | 0 |
| ATM | chr11:108253834:AAAG/A | rs876659575 | E642del | All(2) > 0.44 | 2 | 0 | Both>99% | 0 | 0 |
| MSH6 | chr2:47806344:G/A | rs367912290 | R961C | 0.431818 | 1 | 0 | >99% | 0 | 0 |
| PALB2 | chr16:23626343:C/T | rs766315705 | G586S | 0.571429 | 1 | 0 | 98% | 0 | 0 |
| POLE | chr12:132642888:C/T | rs143247306 | E1554K | 0.516129 | 1 | 0 | >99% | 0 | 0 |
| | chr12:132648934:G/A | rs5744904 | R1355C | 0.454545 | 1 | 0 | >99% | 0.0102041 | 0 |
| | chr12:132649341:G/A | rs779464847 | R1297C | All(2) > 0.57 | 2 | 0 | Both>99% | 0.0102041 | 0 |
| | chr12:132664038:G/C | rs2042735587 | S864C | 0.38 | 1 | 0 | >99% | 0 | 0 |
| | chr12:132672668:A/G | rs115558715 | S549P | 0.447368 | 1 | 0 | >99% | 0 | 0 |
| | chr12:132680212:G/A | rs5744739 | P99L | All(3) > 0.52 | 3 | 0 | >99% (2),97% | 0 | 0 |
| LRP1B | chr2:140442513:C/T | rs144998818 | A3469T | All(3) = 0.54 | 3 | 0 | >99% (2),98% | 0.0102041 | 0.0125 |
| | chr2:140475215:T/C | rs1687922196 | Y3183C | 0.531915 | 1 | 0 | >99% | 0 | 0 |
| | chr2:140598725:C/T | unknown | G2367E | 0.560976 | 1 | 0 | >99% | 0 | 0 |
| | chr2:140841014:G/A | rs199519370 | T1673M | 0.459459 | 1 | 0 | >99% | 0 | 0 |
| | chr2:140886324:C/T | rs752553135 | A1260T | 0.584906 | 1 | 0 | >99% | 0 | 0 |
| ROS1 | chr6:117319878:A/G | rs145889490 | V1977A | 0.536585 | 1 | 0 | >99% | 0 | 0 |
| | chr6:117337265:G/T | rs369993254 | L1719I | ND | 0 | 1 | 78% | 0 | 0 |
| | chr6:117342504:G/T | rs112739824 | P1522Q | 0.666667 | 1 | 0 | >99% | 0.0102041 | 0 |
| | chr6:117365621:C/A | rs370129182 | G978V | All(3) > 0.51 | 3 | 0 | All>99% | 0 | 0 |
| RET | chr10:43120084:G/A | rs145170911 | V871I | 0.617647 | 1 | 0 | >99% | 0 | 0 |
| | chr10:43121991:C/G | rs774215008 | H926D | 0.454545 | 1 | 0 | 98% | 0 | 0 |
| | chr10:43123801:G/A | rs758800351 | E978K | ND | 0 | 1 | 67% | 0 | 0 |
| JAK2 | chr9:5078384:A/C | rs151160183 | N691H | 0.45 | 0 | 1 | 85% | 0 | 0 |
| | chr9:5090448:C/T | rs764302764 | R922W | 0.509091 | 1 | 0 | >99% | 0.0204082 | 0 |
| MSH3 | chr5:80873226:G/A | rs1328941442 | Splice variant | All(3) > 0.44 | 3 | 0 | All>99% | 0 | 0 |
| EGFR | chr7:55142300:A/G | unknown | S35G | 0.55 | 1 | 0 | >99% | 0 | 0 |
| TRRAP | chr7:98976168:C/G | rs143477790 | A2335G | All(3) > 0.47 | 3 | 0 | >99% | 0 | 0 |
| ERBB3 | chr12:56094550:C/G | rs757518347 | T618S | All(3) > 0.38 | 3 | 0 | All>99% | 0 | 0 |

*AA* amino acid, *AFR* African, *chr* chromosome, *EAC* East African Control, *MAF* minor allele frequency, *nt* nucleotide, *PPCG* Pan Prostate Cancer Group, *SAC* Southern African Control, *SAPCS* Southern African Prostate Cancer Study, *VAF* variant allele frequency.

[a]Unknown rs-numbers are absent from variant databases.

[b]Exclusion based on CHIP-likelihood (VAF < 0.3).

[c]African ancestral fraction presented as a percentage (southern, western and/or San) and *k* = 4 ADMIXTURE plot (Fig. 1).

[d]MAF for population-relevant controls. Known PCa germline testing panel genes include *BRCA2, PMS2, FANCA, BARD1, ATM, MSH6* and *PALB2*.

genes *POLQ* (p.Leu232Ile) and *CREBBP* (p.Gln2204 frameshift) warrant further consideration.

Additionally, clonal haematopoiesis of indeterminate potential (CHIP), the natural process of acquiring somatic alterations in haematopoietic stem cells as a person ages, was further considered. After visual confirmation using Integrative Genomics Viewer (IGV)[30], read count was used to determine variant allele frequencies (VAFs) and in turn associated CHIP. Of our 81 rare/PSLF PPV/POV derived candidate genes, five are recognised as CHIP associated[31] and include by ranking

*DNMT3A* (1st), *TET2* (2nd), *PPM1D* (4th), *TP53* (5th), and *JAK2* (7th). Falling within the CHIP associated VAF threshold, defined conservatively here as <0.3[32], all six *DNMT3A* POVs (VAF range 0.205882 to 0.257143), the single *TP53* PPV (VAF = 0.209302) and one each of the three *TET2* POVs (VAF = 0.24) and of the two *PPM1D* POVs (VAF = 0.17) were removed from further analysis. Appreciating missing PPCG VAF data, additional unknown CHIP gene-associated variants removed included both *TRRAP* PPVs occurring in a single 84-year-old patient, the *BRCA2* p.Lys2740X PPV, the *KMT2C* p.Gly3170Ala POV and the single *NCM8*

**Table 3 | Population-Specific Low-Frequency (PSLF) Potentially Pathogenic Variants or Potentially Oncogenic Variants (PPV/POVs, $n=17$) identified in 217 African ancestral prostate cancer (PCa) patients from the SAPCS ($n=186$) and PPCG ($n=31$) study cohorts and impacting 12 DNA Damage Repair (DDR) or PCa-associated genes, either known or unknown as PCa germline testing (GT) gene candidates**

| Gene | PPV/POV | Chr position: nt change | rsID | AA Change | SAPCS (n) | PPCG (n) | MAF (this study) | MAF ALL gnomAD | MAF AFR gnomAD | MAF SAC (n = 49) | MAF EAC (n = 40) | PC | VAF |
|---|---|---|---|---|---|---|---|---|---|---|---|---|---|
| ATM [a] | POV | chr11:108227834:A/G | rs150143957 | D44G | 6 | 0 | 0.0161 (SAPCS) | 0.0002 | 0.0007 | 0.0204 | 0.025 | No | NA |
| PMS2 [a] | POV | chr7:5978683:TGA/T | rs587779335 | L729fs | 2 | 2 | 0.0092 (ALL) 0.0054 (SAPCS) 0.0323 (PPCG) | 0.0077 | 0.0304 | 0.0306 | 0.025 | No | NA |
| PREX2 | POV | chr8:68093713:A/G | rs138027402 | K787E | 6 | 0 | 0.0161 (SAPCS) | 0.0004 | 0.0014 | 0 | 0 | YES | All(6) > 0.46 |
| PREX2 | POV | chr8:68099845:T/C | rs150773140 | Splice | 5 | 0 | 0.0134 (SAPCS) | 0.0006 | 0.0021 | 0 | 0 | YES | All(5) > 0.3 |
| PREX2 | POV | chr8:68121013:C/T | rs143386950 | R1230W | 5 | 0 | 0.0134 (SAPCS) | 0.0000 | 0.0002 | 0 | 0 | YES | All(5) > 0.36 |
| POLG | PPV | chr15:89323423:A/G | rs202037973 | F749S | 5 | 0 | 0.0134 (SAPCS) | 0.0002 | 0.0009 | 0 | 0 | YES | All(5) > 0.42 |
| POLQ | POV | chr3:121537146:G/T | rs141125457 | L232I | 9 | 1 | 0.0230 (ALL) 0.0242 (SAPCS) 0.0161 (PPCG) | 0.0031 | 0.0107 | 0 | 0 | YES | All(10) > 0.39 |
| CREBBP [b] | POV | chr16:3728437:G/GC | rs754959530 | Q2204fs | 4 | 0 | 0.0108 (SAPCS) | 0.0008 | 0.0028 | 0 | 0 | YES | All(4) > 0.46 |
| CREBBP [b] | POV | chr16:3729070: CTCCCGGGG/C | rs777318563 | TPG1990-1992del | 4 | 0 | 0.0108 (SAPCS) | 0.0000 | 0.0003 | 0.0204 | 0 | No | NA |
| FAT1 | POV | chr4:186603302:C/A | rs199719692 | D3742Y | 7 | 0 | 0.0188 (SAPCS) | 0.0002 | 0.0007 | 0.0204 | 0 | No | NA |
| FAT1 | POV | chr4:186663530:C/T | rs149295542 | V117M | 5 | 1 | 0.0138 (ALL) 0.0161 (SAPCS) 0.0161 (PPCG) | 0.0018 | 0.0067 | 0.0204 | 0.025 | No | NA |
| ROS1 | POV | chr6:117352995:A/G | rs17079086 | F1439S | 14 | 0 | 0.0403 (SAPCS) [c] | 0.0033 | 0.0133 | 0.0816 | 0.025 | No | NA |
| ROS1 | POV | chr6:117360384:C/T | rs151330473 | G1135R | 15 | 0 | 0.0403 (SAPCS) | 0.0001 | 0.0008 | 0.0102 | 0.0125 | No | NA |
| ASPM | POV | chr1:197100727:G/A | rs112946633 | R2842W | 23 | 0 | 0.0618 (SAPCS) | 0.0001 | 0.0004 | 0.0306 | 0 | No | NA |
| HERC2 | POV | chr15:28238605: AGAATACTATTTC/A | rs774838885 | GNSIL1245-1249V | 4 | 0 | 0.0108 (SAPCS) | 0.0000 | 0.0003 | 0.0204 | 0.0125 | No | NA |
| PSM1 | POV | chr2:18979188:G/C | rs5742973 | E27Q | 4 | 0 | 0.0108 (SAPCS) | 0.0005 | 0.0014 | 0 | 0.0125 | No | NA |
| TET2 | POV | chr4:105233964:C/G | rs147112198 | H8D | 5 | 0 | 0.0134 (SAPCS) | 0.0006 | 0.0023 | 0.0 | 0 | No | NA |

AA amino acid, AFR African, ALL all populations, chr chromosome, EAC east African Controls, nt nucleotide, MAF minor allele frequency, PC potential candidates, PPCG Pan Prostate Cancer Group, SAC southern African Controls, SAPCS Southern African Prostate Cancer Study, unk unknown.

[a] Known PCa germline testing panel gene.

[b] An additional GT gene candidate not identified within the rare variant analyses (Table 1 and Table 2).

[c] A single patient presented as homozygous.

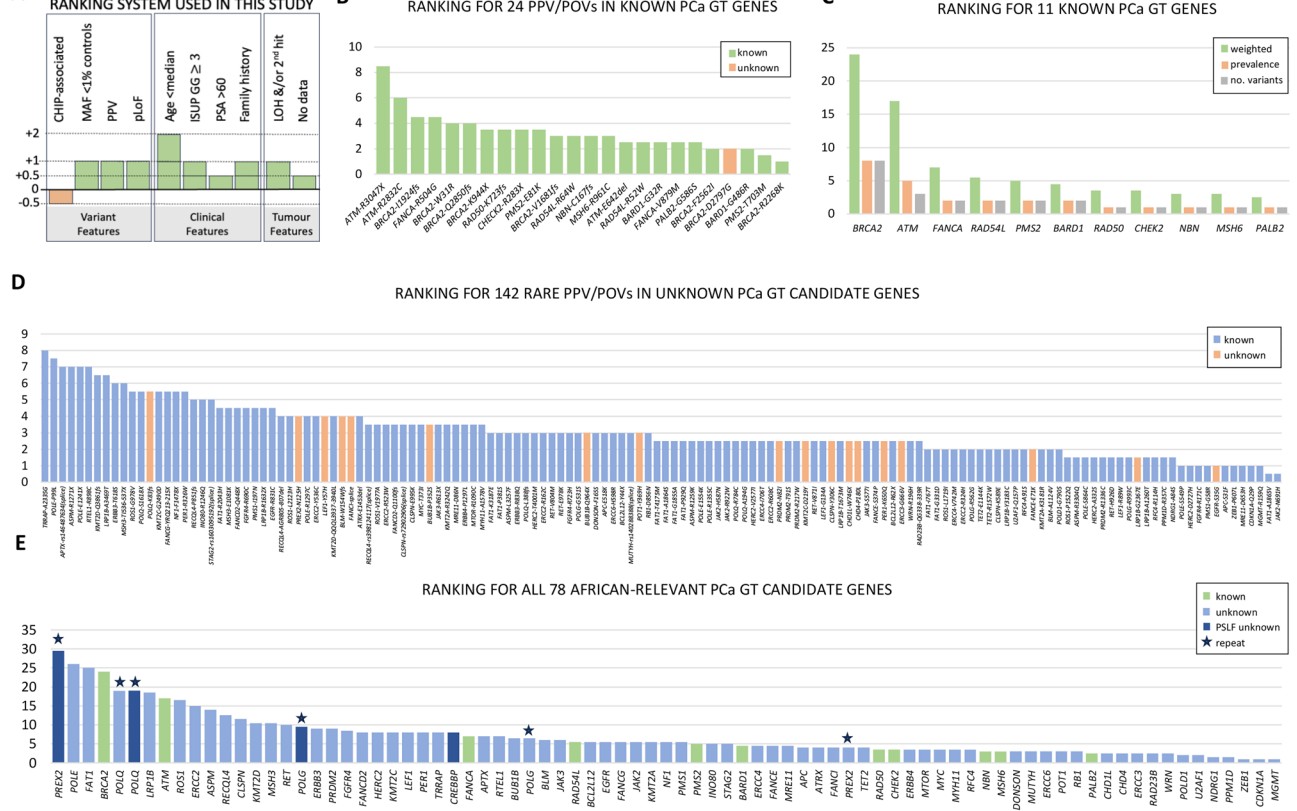

**Fig. 3 | Ranking for potentially pathogenic or oncogenic variants (PPV/POVs) and associated candidate genes for African-inclusive prostate cancer (PCa) germline testing (GT). A** Ranking system overview based on variant, clinical and tumour features. **B** Ranking for 24 rare PPV/POVs identified in known PCa GT genes, including previously reported (known) and not reported (unknown) variants. **C** The 11 known PCa GT genes ranked by weight (total ranked score), prevalence and total number of variants. **D** Ranking for 142 reported (known) and not reported (unknown) rare PPV/POVs impacting 66 candidate genes not included in PCa GT panels. **E** Ranking by weight (ranked score) for all 78 known and unknown PCa GT gene candidates, with population-specific low-frequency (PSLF) candidates assessed independently and represented as gene duplicates (stars), while providing an additional gene candidate *CREBBP*.

POV p.Phe274Ile. After MAF (SAC and EAC) and VAF (CHIP) filtering 41 rare PPVs (32 genes) and 125 rare POVs (59 genes) remained. As none of the PSLF-PPV/POVs fell below the CHIP-associated VAF threshold, all 6 MAF-filtered PSLF variants remained (4 genes). A total of 172 pathogenic variants impacting 78 candidate genes were further considered (Supplementary Data 9).

### Ranking variant pathogenicity and gene candidates

Providing further evidence for our focus on DDR-relevant genes, gene ontology (GO) enrichment and pathway analysis using g:profiler[33] for all 473 genes harbouring low-frequency inclusive PPVs ($n = 252$) and POVs ($n = 529$) revealed DNA damage response and DNA repair as the most enriched biological processes (Fig. S1). Molecular functions were biased towards catalytic activity on DNA and ATP-dependent activity on DNA across the genes. To provide further pathogenic-value to the 172 variants across 78 genes, we developed a 9-step ranking system which provides a weighting (see Methods) for variant features, clinical presentation and when available (116 SAPCS, 31 PPCG) somatic biallelic inactivation (Fig. 3A). A half rank was removed for variants within CHIP-associated genes[31], although well above the VAFs CHIP-threshold, while a full rank was gained for SAC/EAC MAFs <1%, PPV over POV status, and for variants showing potential Loss of Function (pLoF) as estimated using LOFTEE[23]. For clinical features at presentation, less weighting (half a rank) was applied for PSA levels, as elevated non-age-driven PSA heterogeneity has been observed for SAPCS men presenting both with and without PCa[21]. While presenting up to 10 years younger than the study mean, having an ISUP GG ≥4 and a family history of PCa all earned a full rank each, this was doubled for men

presenting over 10 years younger than the study mean and halves for men with a family history of breast or ovarian cancer. Tumour features were defined by loss of heterozygosity (LOH), requiring overlapping somatic copy number loss or somatic SNV with allelic fractions >65% or 15% greater than the germline allele frequency[34], and/or a second hit following Knudson's two-hit hypothesis[35], while a minimal value was applied for missing data (no matched tumour). While our system provides weighting for variance recurrence, gene-matched rare and PSLF PPV/POVs were ranked separately.

Focusing on known PCa GT genes (24 rare PPV/POVs in 11 genes), we observe a study prevalence of 11.06% (24/217), with a single PPCG patient (PPCG0019, 57% African genetic ancestry) presenting with three candidate variants in *RAD54L*, *PMS2* and *FANCA* each, with the latter variant showing a 2nd hit and LOH in the patient-matched tumour. The skewing towards PPCG (25.81%, 8/31) over SAPCS patients (8.60%, 16/186), likely reflects not only the elevated non-African ancestral fractions within PPCG patients, but also the under-representation of southern Africans in PCa genetic data. The highest ranked variants include *ATM* (p.Arg3047X and p.Arg2832Cys), *BRCA2* (p.Ile1924fs), *FANCA* (p.Arg504Gly) and *BRCA2* (p.Trp31Arg and p.Gln2850fs) (Fig. 3B), which includes the highest ranked genes *BRCA2*, *ATM*, and *FANCA*, followed by *RAD54L* and *PMS2* (Fig. 3C). For the unknown gene candidates (142 rare PPV/POVs in 66 genes), the highest ranked variants (>5.5 median) include *TRRAP* (p.Ala2335Gly), *POLE* (p.Pro99Leu), *APTX* (rs146487634 splice donor variant), *ASPM* (p.Arg1271X), *POLE* (p.Glu1241X), *RTEL1* (p.Arg898Cys), *KMT2D* (p.Gln3861fs), *LRP1B* (p.Ala3469Thr), *ERBB3* (p.Thr618Ser), and *MSH3* (p.Ile537fs) (Fig. 3D). While *TRRAP*, *POLE*, *APTX*, *RTEL1* and *MSH3* are known DDR genes,

**Table 4 | SAPCS patients presenting with DNA polymerase PPV/POVs (n = 20) ranked by patient-matched tumour mutational burden (TMB, highest to lowest) and including evidence for microsatellite instability (MSI)**

| POL Gene | AA Change | PPV/POV | Patient ID | TMB | MSS/MSI |
|---|---|---|---|---|---|
| *POLE* | P99L | POV | UP2113[R1] | 59.61010363 | MSI-H |
| *POLQ* | L232I | PSLF-POV | UP2113[R2] | 59.61010363 | MSI-H |
| *POLE* | R1297C | POV | KAL070 | 3.31444300 | MSS |
| *POLE* | P99L | POV | UP2050[R1] | 3.00356217 | MSS |
| *POLG* | R562G | POV | UP2039 | 2.56088082 | MSS |
| *POLQ* | L232I | PSLF-POV | UP2116[R2] | 2.10200777 | MSS |
| *POLE* | E1554K | POV | N0067 | 2.05246114 | MSS |
| *POLQ* | R784C | POV | SMU030[R3] | 1.87726683 | MSS |
| *POLQ* | R784C | POV | N0053[R3] | 1.84455958 | MSS |
| *POLE* | S864C | POV | KAL0074 | 1.59844559 | MSS |
| *POLG* | R993C | POV | KAL0074 | 1.59844559 | MSS |
| *POLQ* | S1618X | POV | SMU050 | 1.53076424 | MSS |
| *POLE* | P99L | POV | UP2197[R1] | 1.14831606 | MSS |
| *POLQ* | L232I | PSLF-POV | UP2004[R2] | 1.02040155 | MSS |
| *POLQ* | L388del | POV | UP2004 | 1.02040155 | MSS |
| *POLQ* | K83fs | POV | KAL0106 | 1.00680051 | MSS |
| *POLQ* | L232I | PSLF-POV | SMU097[R2] | 0.76878238 | MSS |
| *POLG* | F749S | PSLF-PPV | SMU177[R4] | 0.46599741 | MSS |
| *POLG* | G531S | POV | UP2317 | 0.37240932 | MSS |
| *POLG* | F749S | PSLF-PPV | KAL0101[R4] | 0.07059585 | MSS |
| *POLG* | F749S | PSLF-PPV | SMU159[R4] | 0.0625 | MSS |
| *POLE* | R1355C | POV | SMU111 | 0.0625 | MSS |
| *POLQ* | L232I | PSLF-POV | UP2092[R2] | 0.03465025 | MSS |

*AA* amino acid, *PPV* potentially pathogenic variant, *POV* potentially oncogenic variant, *PSLF* population specific low-frequency, *TMB* tumour mutational burden, *MSS* microsatellite stability, *MSI-H* microsatellite instability high.
[R1-R4]Represent patients (by ID) presenting with the same POL gene pathogenic variant.

**Table 5 | SAPCS patients (n = 22) presenting with potentially pathogenic germline variants (PPV/POVs) and showing tumour-matched enrichment for DNA damage repair (DDR)-like mutational signatures**

| Gene | AA change | PPV/POV | Ranking | Patient Tumour ID | Mutational Signature |
|---|---|---|---|---|---|
| *BRCA2* | W31R | PPV | 4 | UP2103 | ID6 |
| *PREX2* | R1230W | PSLF-POV | 8 | | |
| *FANCA* | R504G | POV | 4.5 | N0084 | ID1 |
| *FANCG* | YRQ213-215X | PPV | 5.5 | UP2093 | DSB7 |
| *PREX2* | R1230W | PSLF-POV | 8 | | |
| *RECQL4* | AGR805-807del | PPV | 4 | UP2187 | DSB7 |
| *CLSPN* | Splice donor | POV | 3.5 | | |
| *ERCC4* | I706T | POV | 2.5 | N0078 | ID1 |
| *RAD23B* | QG338-339R | POV | 2.5 | | |
| *POLE* | P99L | POV | 7.5 | UP2113 | ID2 |
| *POLQ* | L232I | LF-POV | 19 | | |
| *FAT1* | A1865V | POV | 0.5 | SMU030 | ID1 & SV3 |
| *POLQ* | R784C | POV | 2.5 | | |
| *HERC2* | A332S | POV | 1.5 | KAL0091 | SV3 |
| *LEF1* | R89W | POV | 1.5 | | |
| *PREX2* | R1230W | PSLF-POV | 8 | | |
| *RET* | V871I | POV | 2.5 | SMU041 | SV3 |
| *KMT2A* | R3242Q | POV | 3.5 | | |
| *MUTYH* | splice donor | POV | 3 | SMU097 | SV3 |
| *POLQ* | L232I | PSLF-POV | 19 | | |
| *FANCI* | splice donor | POV | 4 | UP2330 | ID2 |
| *TRRAP* | A2335G | POV | 8 | UP2133 | ID1 |
| *RTEL1* | R898C | POV | 7 | UP2258 | ID1 |
| *LRP1B* | Y3183C | POV | 2 | UP2396 | ID6 |
| *ERBB3* | R838Q | PPV | 3 | N0007 | ID8 |
| *POLQ* | S1618X | POV | 5.5 | SMU050 | ID1 & SV3 |
| *FAT1* | G331D | POV | 2 | UP2372 | SV3 |
| *ERCC4* | A860D | POV | MAF excluded | SMU167 | SV3 |
| *KMT2A* | K3181R | POV | 2 | KAL0078 | SV3 |
| *PREX2* | K787E | PSLF-POV | 9 | N0001 | SV3 |
| *ERCC2* | R608C | POV | 2.5 | N0088 | SV3 |
| *DNMT3A* | R326P | POV | VAF excluded | SMU083 | SV3 |

*AA* amino acid, *PPV* potentially pathogenic variant, *POV* potentially oncogenic variant, *PSLF* population specific low-frequency, *ID* insertion-deletion, *DBS* double-base-substitution, *SV* structural-variation, *CN* copy-number.

more recently *ASPM*[36], *KMT2D*[37], *LRP1B*[38] and *ERBB3*[39] have been defined as DDR relevant. Merged with our known and PSLF gene candidates, while *PREX2* PSLF variant restricted, *POLE* and *FAT1* outrank *BRCA2*, and *POLQ* and *LRP1B* outrank *ATM* (Fig. 3E). When combining rare and PSLF variants, *POLQ* outranks *PREX2*, while *POLG* ranking approaches that of *ATM*. In contrast to the DDR DNA poloymerase genes, *POLE*, *POLQ* and *POLG*, and DDR-relevant genes, *PREX2* and *LRP1B*, *FAT1* is a known PCa tumour suppressor[40]. Additional unknown candidate genes outranking *FANCA* include known DDR genes *ERCC2*, *RECQL4*, *CLSPN*, *MSH3*, *FANCD2*, *HERC2*, *TRRAP* and *CREBBP* (PSLP driven), DDR-relevant genes *ROS1*, *ASPM*, *KMT2D*, *ERBB3*, *PRDM2*, *FGFR4*, *KMT2C*, *LEF1* and *PER1*, and the PCa germline associated oncogene *RET* (Supplementary Data 10).

## Southern African patient-matched tumour mutational burden and signatures

Besides tumour features linked directly to PPV/POV ranking, having observed an overall higher tumour mutational burden (TMB, 1.197 *vs* 1.061 mutations/Mb, Log10-transformed $t = 2.5207$, $P = 0.01308$) and enrichment of mutational signatures of unknown significance (10 vs 1) in our SAPCS *versus* European-derived tumours[18], we further sought to

correlate biologically relevant PPV/POV status with patient-matched TMB, with a focus on the PPV/POVs impacting the DNA polymerases, and tumour enrichment for signatures known to be associated with the same or similar largely DDR-related aetiologies. Ranking TMBs for all 116 SAPCS patients, 10/20 (50%) of DNA polymerase presenting PPV/POV patients presented with a TMB above the median (1.23 mutations/Mb), ranging from 1.53 to 3.31 mutations/Mb and including a single

outlier UP2113 (59.61) with associated microsatellite instability (MSI) (Table 4). Three patients presented with two POL gene PPV/POVs each, including the TMB outlier (*POLE* p.Pro99Leu and *POLQ* p.Leu232Ile), while KAL0074 (*POLE* p.Ser864Cys and *POLG* p.Arg993Cys) presented with an above median TMB (1.598). Notably, mutational signatures associated with TMB or DNA polymerase variants, such as single-base-substitution (SBS)9, SBS10 (all), SBS14 and double-base-substitution (DBS)3, were absent in our study.

While the *BRCA2*-associated signature SBS3 was found to be enriched in a single SAPCS patient with no DDR/known PCa-associated PPV/POV germline variant, no enrichment was observed for signatures with associated DDR-related aetiologies, including SBS6, SBS15, SBS21, SBS26 and SBS44, while the *MSH6* POV carrier did not present with the gene-associated copy-number (CN)25 tumour enrichment. Conversely, 22 PPV/POV-presenting SAPCS patients harboured DDR-like mutational signatures (Table 5), including DBS7 (defective DNA mismatch repair), insertion-deletion (ID)1 and ID2 (defective DNA mismatch repair/DNA replication slippage), ID6 (homologous recombination DNA damage repair associated with *BRCA2/1* mutations), ID8 (repair of DNA double strand breaks by non-homologous DNA end-joining mechanisms) and structural-variation (SV)3 (homologous recombination deficiency), of which 9/22 (40.9%) or 9/20 (45%, excluding for MAF/VAF criteria) presented with two or more PPV/POVs. Notably, two patients with *POLQ* POVs (p.Arg784Cys and p.Ser1618X) showed enrichment for both ID1 and SV3.

## Discussion

Recent research indicates that 88% of early PCa mortality occurs in individuals with high genetic susceptibility or a family history of cancer, while only one-third of these deaths are preventable through lifestyle modification[41]. Additionally, outcomes for patients with DDR-specific pathogenic variants have been shown to ameliorate with adjunct hormone therapy or chemotherapy[2], including a positive response to poly-(ADP ribose) polymerase (PARP) inhibitors[42]. Taken together, this underscores the importance of GT, which is gaining momentum[4]. Targeting largely DDR genes, the prevalence among men meeting NCCN screening criteria is estimated at 15–17%[11,12]. Focusing on 60 cancer susceptibility genes, a recent study of 1883 men undergoing tumour WGS, irrespective of clinical presentation yet biased towards metastatic disease, found 22% with a cancer driver also presented with an actionable pathogenic germline variant[43]. As with the latter study, current literature has almost exclusively focused on European ancestral populations. As such, detecting pathogenic variants in African populations at greatest risk for PCa-associated mortality is hindered by a paucity of data[10,14].

Here we perform a comprehensive non-targeted WGS-based interrogation for African ancestral PCa patients, with a focus on the region most impacted by associated lethality−southern Africa[15]. Reporting a prevalence of 5.99% for PPVs in known PCa GT candidate genes (12 PPVs, 6 genes in 13 patients), restricting our analysis to men with > 90% African genetic ancestry reduced the prevalence to 4.69% (9/192) and a roughly 3-fold reduction in reported PCa GT efficiency. Appreciating that African-relevant PPVs are likely underrepresented in current databases, exacerbated by European-centric guidelines, we used a previously employed method to filter VUS with a high possibility of oncogenicity[10]. Identifying 12 POVs in 12 patients, we increased the number of represented known PCa GT genes to 11 and a prevalence of 11.06% (24/217 all African) or 9.90% (19/192 restricted African ancestry >90%), which remains below that reported for non-African populations. While the most impactful variants defined by our ranking system were both in *ATM* (p.Arg3047X and p.Arg2832Cys), overall the most impacted known PCa GT gene was *BRCA2*. Conversely, no PPVs/POVs were identified in *BRCA1, HOXB13, CDK12, MLH1, MSH2*, or *BRIP1*.

The decreased prevalence for known PCa GT candidate-impacted genes in our African cohort, with further genetic conservation of six candidates, further highlights the potential for yet unknown African-inclusive gene candidates. Irrespective of gene candidates or function, we found notable enrichment for DDR biological processes for genome-wide PPV/POVs, providing further justification for tailored gene discovery. Aware of the under-representation of African-derived data in ClinVar and used for the development of ACMG/AMP guidelines, it was essential that we provide further clarification for VUS, which, taken together, resulted in the identification of 148 rare/PSLF PPV/POVs across 67 unknown gene candidates. Notably, *PREX2, POLE* and *FAT1* outrank *BRCA2*, while *POLQ* and *LRP1B* outranks *ATM*. Overall, the DNA polymerases *POLE, POLQ*, and *POLG* represent the highest combined rankings, with the latter two including PSLF-POV representation. This coincides with a recent study reporting germline *POLE* and *POLQ* variants in African American PCa patients[44], while the reported benefit for Durvalumab therapy in colorectal cancer patients with germline *POLE* mutations[45] holds potential for PCa precision oncology. Additionally, we found 50% of the tumour-matched SAPCS *POLE, POLQ*, and *POLG* carriers to present with an above median TMB. While Fanconi Anemia-associated genes *BRCA2, FANCA* and *PALB2* are known PCa GT candidates[11,12], *FANCD2* outranked *FANCA*, with *FANCG, ECCR4, FANCE* and *FANCI* (in order of ranking) potential candidates. Intriguingly, the *FANCG* p.Tyr213fs deletion has previously been associated with breast cancer in a South African patient[46]. While DNA mismatch repair genes *MSH6* and *PMS2* are known PCa GT candidates[11,12], unknown candidates *MSH3* and *PMS1* out-ranked their namesake counterparts by 3.5- and 1.1-fold, respectively. Our findings are further supported by *MSH3* germline rare variants having been associated with PCa in Chinese patients[47], while rare *PMS1* variant has been linked to hereditary breast cancer[48]. Two of the three DNA helicase genes *RECQL4* and *BLM* rank 7 and 0.5 points above the study median, respectively, with *RECQL4* supported by published PCa germline variants[49,50]. We found *KMT2D, KMT2C, TRRAP* and *CREBBP*, genes involved in chromatin remodelling, to outrank *FANCA*. Conversely, the epigenetic modulators *DNMT3A* and *TET2* showed CHIP-associated VAFs for all six *DNMT3A* one of three *TET2* variants. While *DNMT3A* was removed from our candidate gene list, rare *TET2* variants have been reported for African American PCa patients[51]. Additionally, while the single PPV in the known PCa GT and highly ranked CHIP-associated gene *TP53* showed evidence for non-inheritance and was as such removed, all three PPV/POVs in the highly ranked CHIP-associated gene *JAK2* were retained as somatic, achieving a median ranking. Another Janus kinase (JAK) gene making the list included *JAK3* (6 ranking).

Providing insights for possible African-relevant PCa GT candidate genes, it is notable that although a recent DDR-targeted study of 17,000 European PCa patients advocated for the inclusion of *XRCC2, MRE11, POLK, POLH*, and *MSH5*[9], only *MRE11* (4.5 ranking) was identified in our study. Irrespective of ancestry, however, both studies call for focus on the DNA polymerase genes. Additionally, while NCCN guidelines[2] recommend the inclusion of *BARD1* (4.5 ranking) and *RAD54L* (5.5 ranking), these genes are largely absent from commercially available panels[12]. Besides the missense POVs reported here, recently we described a *BARD1* pLoF large deletion in a SAPCS patient with associated somatic LOH[52], emphasising the potential for overlooked inherited structural variants through our study focus on small variants. Other potential limitations include assessing for pLoF in oncogenic candidates such as *RET, ROS1, FGFR4*, and *MYC*, while *FAT1* and *LEF1* reported to oscillate between oncogenic and tumour suppressive behaviour. While no PPV/POVs identified in these genes showed pLoF, we are unable to determine their potential gain-of-function. Additionally, *ROS1* (ranking 11 points above the median) has been shown to display DDR activity[53], is a *BRCA*-negative breast cancer gene candidate[54], and has been shown to harbour PPVs in Chinese PCa

patients[55]. Furthermore, our highest impacted gene *PREX2*, a DDR-relevant oncogene[29], harboured a single splice donor disrupting pLoF PSLF-POV requiring further functional clarification. While our data alludes to the benefits of our whole genome approach, we acknowledge limitations of defining true functionality, with the inevitable potential for pathogenic misclassification. Additionally, while candidate PPV/POVs in known GT genes, including *BRCA2* (p.Trp31Arg) and *FANCA* (p.Arg504Gly) showed tumour associated DDR-like mutational signature enrichment, the 28 PPV/POVs in unknown GT-candidates showing DDR-like mutational enrichment provides further merit for consideration.

Besides unknown and overlooked gene candidates, the lack of guidelines or management plans for over 20% of current GT genes identified in PCa has limited GT application[56]. Increased affordability and accessibility for GT have seen a growth in uptake among men not meeting NCCN criteria[57], with a caveat of poor panel coverage leading to negative results and false reassurance[11], which is more likely in African populations who exhibit understudied and distinct genetic patterns[8,10,18,28]. Noting that numerous actionable germline variants are overlooked using current panels, a recent non-African study advocated for WGS as a cost-effective alternative[58]. Additional non-genomic considerations include the elevated clinical heterogeneity observed across ethno-linguistic groups from the same region within sub-Saharan Africa[59,60], while defining high- or very-high-risk PCa based on European-derived NCCN PSA inclusion criteria (PSA > 20 ng/mL) for PCa GT screening, as shown for SAPCS[21], requires African-specific criteria. In concordance with others[6,61,62], we need to consider reduced PCa awareness in addition to cultural barriers driving later diagnosis and reduction in knowledge with regards to family history as observed for more rurally located SAPCS recruits[17,63].

In conclusion, our findings underscore the complexity of designing an African-inclusive GT panel for PCa, necessitating multiple panels or a broader range of genes than those pertinent to non-African populations. Our refined set of genes and germline variants provides a much-needed framework for stratification in clinical trials and serves as a roadmap for functional validation studies. These can be utilised across African populations in precision medicine, with potential applications extending both within Africa and worldwide.

# Methods

## Ethics and inclusion statement

As per the HEROIC PCaPH Africa1K inter-institutional Collaborative Research Agreement (CRA) and Global Code of Conduct for research in resource-poor settings, locals have been included in all aspects of the research including study design, local primary ethics approvals and stewardship, study implementation, analysis and authorship, to intellectual property and data ownership. Capacity building across South Africa and Kenya includes (i) awarded and self-managed budget allocation, which has led to numerous employments including clinicians, scientists, nurses, field workers and administrators, (ii) sourcing infrastructure, resourcing and providing clinical training to provide much needed urology screening in under-resourced regions, (iii) co-supervision and exchanges for postgraduate students to genomic intensive partner laboratories, (iv) providing access to off-site high performance computational infrastructure, while (v) holding on-site annual training workshops in projects related topics. Through engagement and inclusion of local policy makers, consumer representatives and public health leaders, the team is committed to the dissemination of scientific data back to communities and local government.

## Ethics approvals and institutional agreements

Biological male patients (verification of prostate organ) and population-representative sex/gender-unbiased controls provided informed consent to participate in the study and were recruited as part of the SAPCS (patients and controls) or East African Prostate Cancer Study (EAPCS, controls only). For the SAPCS, study approval was granted by the University of Pretoria Faculty of Human Research Ethics Committee (HREC #43/2010, including US Federal-wide Assurance FWA00002567 and IRB00002235 IORG0001762) in South Africa, with additional Institutional Review Board (IRB) approval granted by the Human Research Protection Office (HRPO) of the US Army Medical Research and Development Command (E02371.2a TARGET Africa; E03333.1a and E05986.1a HEROIC PCaPH Africa1K). For the EAPCS, study approval was granted by the Kenyatta National Hospital (KNH) and University of Nairobi (UON) Ethics Research Committee (ERC) in Kenya (KNH/UON-ERC P637/07/2019), with additional IRB approval granted by the US Army Medical Research and Development Command HRPO (E03347.1b and E05987.1a HEROIC PCaPH Africa1K). Samples (whole blood) were shipped to the University of Sydney in accordance with institutional Material Transfer Agreements (MTAs) and including for the SAPCS under a Republic of South Africa Department of Health Export Permit (National Health Act 2003; J1/2/4/2), while data sharing includes is made possible by a full-executed inter-institutional CRA between the HEROIC PCaPH Africa1K study leads including the University of Sydney (Australia), University of Pretoria (South Africa), University of Nairobi (Kenya) and University of Chicago (U.S.A.). Molecular genetic research for patients from the SAPCS bioresource was approved by the St. Vincent's Hospital Human Research Ethics Committee in Sydney in Australia (#SVH15/227), with additional IRB approval granted by the US Army Medical Research and Development Command HRPO (E02371 TARGET Africa; E03280.1a and E05984.1a HEROIC PCaPH Africa1K). As an International Cancer Genome Consortium (ICGC) member, the PPCG collection is subject to the standards of ethical consent. Country-specific IRB approvals, which included Australian samples from Melbourne (Epworth Health 34506; Melbourne Health 2019.058) and Sydney (St Vincent's HREC #SVH/12/231).

## Participants

**PCa patients.** The 217 African ancestral participants were recruited either at routine, and as such non-compensated, PCa diagnosis from a participating SAPCS urology clinic in South Africa or at radical prostatectomy from a participating PPCG member site. Study inclusion was based on a histopathological confirmation of PCa defined as a Gleason score or an International Society of Urological Pathology Grade Group (ISUP) and a self-reported and/or genetically predicted African ancestry. For the SAPCS, 186 men self-identifying as African ancestral or more specifically from a southern African Bantu ethno-linguistic group, were selected for whole genome interrogation, including both published (n = 116)[18] and unpublished data (n = 70). The additional PCa patients represented South Africans recruited at research hubs for the TARGET Africa and/or HEROIC PCaPH Africa1K US-DoD-funded projects, which included Dr George Mukhari Academic Hospital of the Sefako Makgatho Health Sciences University, an urban hub in the province of Gauteng, or at Tshilidzini Hospital, an approved University of Pretoria research hub, within the rural province of Limpopo. Conversely, the PPCG includes whole genome data for 959 PCa cases sourced from Canada (n = 303), Germany (n = 238), United Kingdom (n = 226)[64,65], Australia (n = 143 Melbourne, 53 Sydney)[18], and France (n = 25), of which 31 (3.1%), including 11 Canadians, 10 British and 10 French Caribbeans, reported African ancestry[18].

**African controls.** The HEROIC PCaPH Africa1K has access to 49 southern Africans self-identified from one or more southern Bantu ethno-linguistic group and recruited as part of the SAPCS, and 40 east Africans self-identified from either an eastern Bantu or Nilotic ethno-linguistic group via the EAPCS. Participation as a population-matched study control included two-generational African ethno-linguistic identity, being less than 50 years of age, no PCa or any cancer

diagnosis, and unlike our case cohort, representing any self-reported gender. Having undergone deep whole genome sequencing (unpublished), provided the background for targeted candidate gene interrogation for population-relevant MAFs.

**Healthy controls.** The MGRB samples were gathered from 3,209 European ancestral Australian individuals aged 75 years or older with no known metabolic illnesses including hypertension, cancer, or dementia[20]. WGS of the samples was performed on Illumina HiSeq X sequencers, generating a median coverage of 37.31X (range 21.95 to 44.12X). Mapping was built on GRCh37 and variant calling was performed following GATK best practices as previously described[20].

## Whole genome sequencing and variant calling

As previously described for the SAPCS[18], DNA was extracted from whole blood (Qiagen kits) from treatment-naïve patients and 2 x 150 cycle paired-end whole genomes were sequenced (Illumina HiSeq X Ten or NovaSeq) to an average of 45X coverage (range, 30 to 71X) and aligned to the GRCh38 reference. SNVs and small insertions and deletions (indels; <50 base pairs) were called using the Genome Analysis Toolkit (GATK v4.1.2.0, Broad Institute)[66] and variant data made available through the SAPCS Data Access Committee (DAC), with data deposited for 116 published genomes at the European Genome-phenome Archive (Table S1). Another 70 Southern African PCa patients were deep sequenced using the Illumina NovaSeq Plus (University of New South Wales Ramaciotti Genomics Facility) to an average of 43.3X coverage (range 36.4 to 69.1X), with SNVs and indels called using the Sydney Informatics Hub quality control (QC) and germline-ShortV joint-calling (see Code Availability). PPCG whole genome data have been generated by each participating country, as previously described[19], with data sourced from Australia, including Sydney's Garvan/St Vincent's PCa Database[18] and Melbourne Research group, Canadian PCa Genome Network, French ICGC PCa group, Germany ICGC PCa group, and CRUK-ICGC Prostate Group, UK. Apart from the Australian Sydney variant data called using the SAPCS pipeline[18], all remaining PPCG variants were called using a single GRCh37-referenced liftover[19].

## Genetic ancestral fractions

Further clarification of African ancestry and population substructure was performed for all 217 cases. Representative control populations from the Human Genome Diversity Project (HGDP) and 1000 Genomes Project (1KGP) and incorporated within the gnomAD v3.1. The database included 20 individuals each representing East African (Luhya, LWK), West African (Yoruba, YRI), African American (ASW), European (CEU) and Asian (Han Chinese, CHB) ancestries[23]. The 20 southern African KhoeSan were derived from the KhoeSan Genome Project (KSGP, unpublished data Hayes Lab). Using a set of 77,369 linkage disequilibrium (LD)-pruned exomic single nucleotide variants (SNVs), previously used to characterise the major substructure between African regions[28], after filtering for variants that were not fixed in the current dataset, a total of 64,654 SNVs were used for ADMIXTURE v1.3.0[67] analysis and tested for $k = 1$ to 10 with five-fold cross-validation (CV) and 10 replications each. While $k = 3$ generated the lowest mean CV error at 0.2525 (10/10 replicates in concordance), $k = 4$ had slightly higher mean CV error at 0.255 (10/10 replicates in concordance) and could distinguish Southern African ancestry from West African ancestry, which was used to further refine patient ancestral population substructure.

## Variant pathogenicity prediction and classification

Following the identification of pathogenic/likely pathogenic variants in the Clinvar database, which includes the American College of Medical Genetics and Genomics and the Association for Molecular Pathology (ACMG/AMP) guidelines, variants with a population minor allele frequency (MAF) < 5%, as defined using gnomAD v.4.0[23], were recorded here as potentially pathogenic variant (PPVs). Genes whose link to DDR was more recently discovered, as well as genes with evidence of reported germline variants in PCa were also included. Genes harbouring PPVs among the African PCa patients were interrogated for variant pathogenicity among the PPCG European PCa patients and the MGRB healthy controls. The genes were excluded from the African-relevant list if the overall MAF of the PPVs were higher in these populations compared with the African patients. For all the remaining variants, those reported as deleterious or damaging using the SIFT[25] and PolyPhen-2[26] prediction tools, respectively, that resulted in a stop codon or splice junction disruption were further selected. Variants were removed if they were reported as benign/likely benign in ClinVar or by the ACMG/AMP guidelines or had an MAF > 5% from all population-defined gnomAD data. Finally, variants were described as potentially oncogenic variant (POVs) if they were reported as an oncogenic driver in the Cancer Genome Interpreter (CGI)[27]. These variants were further refined to include those involved in DDR[24], those with evidence of germline variants in PCa (according to the same standards), and MAF < 1%. All candidate PPVs and POVs were visually confirmed through allele frequencies using Integrative Genomics Viewer (IGV)[30].

## Candidate gene ranking

To confirm that the variants in our candidate gene list were inherited rather than resulting from CHIP, we analysed read counts to ascertain variant allele frequencies, removing variants with VAF < 30%[32]. For our 9-step ranking system, variant feature weighting included (i) CHIP-associated gene (−0.5), (ii) SAC/EAC MAF < 1% (+1), (iii) PPV over POV (+1), and (iv) pLoF (+1), clinical features of patients at diagnosis/surgery with weighting included (v) age up to 10 years younger (+1) or over 10 years younger (+2) than cohort mean (mean 67 years for SAPCS and 65 years for PPCG patients), (vi) ISUP GG = 3 (+0.5) and ≥ 4 (+1), (vii) PSA > 60 ng/mL (+1), which is based on the more conservative PPCG cohort mean, and (viii) family history (1st or 2nd-degree relatives) of PCa (+1) or breast and/or ovarian cancer (+0.5), and lastly (ix) tumour features including gene-matched LOH and/or second somatic hit (+1), while factoring for samples where tumour was not available (+0.5).

## Statistics and reproducibility

Sample size was determined by the availability of recruited patients and/or whole genome data meeting the study criteria, African ancestral patients with a clinicopathological diagnosis of PCa. As such, no statistical method was used to predetermine sample size and after meeting inclusion criteria, no patient/data were excluded from the analyses. While the experiments were not randomised, for both initial SAPCS and PPCG data generation and analyses, investigators were blinded to patient ancestry. After genetic testing, men of confirmed African ancestry were selected for downstream analyses.

## Reporting summary

Further information on research design is available in the Nature Portfolio Reporting Summary linked to this article.

# Data availability

Access to published whole genome sequence data published in Jaratlerdsiri et al[18] was made available via Data Access Committee (DAC) approval as outlined under the European Genome-Phenome Archive (EGA) [https://ega-archive.org] project-specific access policies under overarching study EGAS00001006425, which includes the Southern African Prostate Cancer Study (SAPCS) Dataset at EGAD00001009067 and as part of the PPCG cohort the Garvan/St Vincent's Prostate Cancer or Sydney Database at EGAD00001009066, while additional PPCG Datasets are summarised in Table S1 and include Canadian PCa Genome

Network [https://ega-archive.org/datasets/EGAD00001004170], CRUK-ICGC Prostate Group UK [https://ega-archive.org/datasets/EGAC00001000852], French/Caribbean ICGC PCa Group [https://ega-archive.org/datasets/EGAD00001003835], Germany ICGC PCa Group [https://ega-archive.org/datasets/EGAD00001005997], and Melbourne Research Group Australia [https://ega-archive.org/datasets/EGAD00001004182]. MGRB data is available as defined by study EGAS00001003511 and dataset EGAD00001005228. The additional 70 SAPCS germline whole genome data has been deposited under the overarching study EGAS50000001132 [https://submission.ega-archive.org/submissions/EGA50000001053] and dataset EGAD50000001626 [https://submission.ega-archive.org/submissions/EGA50000001053/datasets]. Additional variant and annotation data for the African PCa patients, European PPCG patients, African and healthy control populations study are available within the main text and supplementary information.

Access to the additional SAPCS sequencing data generated in this study may be requested via the SAPCS DAC and will be made available to researchers with appropriate feasibility and corresponding ethics approvals to ensure the safeguarding of patient genomic information (contact V.M.H. or M.S.R.B.). Restrictions include (i) No transfer to third parties allowed, (ii) acknowledgment of the SAPCS in publications/presentations, (iii) a report of the results of the research to be provided to DAC after completion (or when requested), (iv) researchers cannot utilise the data for commercial purposes or any other purposes not approved by the DAC, and (v) approval will not be given that excludes other researchers from accessing data. Data currently being used for capacity building in under-resourced studies across Sub-Saharan Africa will be given priority and at times may be granted time-limited exclusive rights for no more than a two-year period.

SNVs and indels data supporting the findings of this study are available within the main text and Supplementary information. Previously published SNV and indel sites and their minor allele frequencies are available in the dbSNP [https://www.ncbi.nlm.nih.gov/snp/][68], and gnomAD databases [https://gnomad.broadinstitute.org/][23]. Gene regions are available in the ENSEMBL database [https://www.ensembl.org][69], and DDR gene list is available at GSEA[24].

## Code availability
Software and scripts for DNA sequence read data collection, and the scripts for sequence read alignment and quality control are available on GitHub (https://github.com/Sydney-Informatics-Hub/Bioinformatics).

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

## Acknowledgements

The authors are forever grateful to the patients, their families, respective clinical staff and bioresource managers who have who have contributed to the data within each Consortium represented in this study. Specifically, we acknowledge the contributions of the SAPCS participants and health care workers from South Africa who have contributed both published[24], and additional data. Furthermore, we thank the Ramaciotti Centre for Genomics, University of New South Wales, Sydney, for data generation, as well as the Sydney Informatics Hub at the University of Sydney and the National Computational Institute (NCI) in Canberra for providing the high-performance computational infrastructure Artemis

and Gaddi used in this study, respectively. We also acknowledge the support of the research staff in the PPCG contributing teams who so carefully curated the samples and the follow-up data. Genomic sequencing and interrogation of SAPCS data was supported by the Ancestry and Health Genomics Laboratory at the University of Sydney through National Health and Medical Research Council (NHMRC) of Australia funding (2018/GNT1165762, 2020/GNT2001098 and 2021/GNT2010551 to V.M.H.), and USA Congressionally Directed Medical Research Programs (CDMRP) Prostate Cancer Research Program (PCRP) funding, including an Idea Development Award (PC200390, TARGET Africa to V.M.H.) and HEROIC Consortium Award (PC210168 and PC230673, HEROIC PCaPH Africa1K to V.M.H., M.S.R.B., P.M.N. and G.S.P.), the latter including data generation for African control data. For PPCG curation, management and analysis (to R.A.E. and Z.K-J.) we acknowledge support from Cancer Research UK (C5047/A14835/A22530/A17528, C309/A11566, C368/A6743, A368/A7990, C14303/A17197), Prostate Cancer UK (MA-TIA23-002), Dallaglio Foundation (CR-UK Prostate Cancer ICGC Project and Pan Prostate Cancer Group), PC-UK/Movember, the NIHR support to The Biomedical Research Centre at The Institute of Cancer Research and The Royal Marsden NHS Foundation Trust. Further support for SAPCS and PPCG analysis was provided by the USA National Institute of Health (NIH) National Cancer Institute (NCI) Award (1R01CA285772-01 to V.M.H.) and a USA Prostate Cancer Foundation (PCF) Challenge Award (2023CHAL4150 to V.M.H., M.S.R.B. and G.S.P.). V.M.H. is further supported by the Petre Foundation via the University of Sydney Foundation (Australia) and M.M.H. by a Sydney Cancer Partners pilot grant (Australia).

## Author contributions

Conception and design: V.M.H. Administrative support: J.J., T.M.N.M., M.T.L., S.M.P., G.S.P. and K.D.S. Provision of study materials or patients (SAPCS and controls): R.A.C., M.B.R., M.N., Muv.O., Mar.O., S.B.A.M., M.S.R.B. and V.M.H., and (EAPCS controls): W.M.O., M.O.O. and P.M.N. Pathology review: Mel.L. and Mas.L. Sample processing and preparation (70 SAPCS genomes): M.M.H. Variant calling and data uploads (70 SAPCS genomes): J.J. Collection and assembly of data: K.G., P.X.Y.S., J.J., D.B., P.M., D.K., W.J., D.C.W., R.G.B., D.S.B., C.S.C., J.R., G.C-T., O.C., C.M.H., N.M.C., P.D.S., T.S., J.W., S.B.A.M., P.M.N., D.M.T., Z.K-J., R.A.E., M.S.R.B. and V.M.H. Ancestry fraction determination: P.X.Y.S. Data analysis and interpretation: K.G., P.X.Y.S., J.J., D.B., P.M., D.K., J.W., D.M.T., Z.K-J., R.A.E. and V.M.H. Manuscript writing and Figure generation: K.G., P.X.Y.S. and V.M.H. Final approval and review of manuscript: All authors

## Competing interests

Member of Active Surveillance Movember Committee (V.M.H., R.A.E.). Member of external expert committee to Astra Zeneca UK (R.A.E.). Honoraria from GU-ASCO, Janssen, University of Chicago, Dana Farber Cancer Institute USA as a speaker (R.A.E.). Educational honorarium from Bayer and Ipsen (R.A.E.). Member of the SAB of Our Future Health (R.A.E.). Undertakes private practice as a sole trader at The Royal Marsden NHS Foundation Trust and 90 Sloane Street SW1X 9PQ and 280 Kings Road SW3 4NX, London, UK (R.A.E.). The remaining authors declare no competing interests.

## Additional information

Kazzem Gheybi [1], Pamela X. Y. Soh [1], Jue Jiang [1], Tumisang M. N. Mbeki[2], Melanie Louw[3,4], Daniel Burns[5], Piyushkumar Mundra[6], Daria Kiriy[7], Md. Mehedi Hasan[1], Weerachai Jaratlerdsiri [1,8], Maphuti Tebogo Lebelo[9], Raymond A. Campbell[10], Mulalo B. Radzuma[11], Mukudeni Nenzhelele[12], Muvhulawa Obida[2,12], Martin Obida[2,12], Winstar M. Ombuki[13], Micah O. Oyaro[14], Sean M. Patrick[2], Massimo Loda [15], David C. Wedge [16], Robert G. Bristow [16], Daniel S. Brewer [17,18], Colin S. Cooper[5,17], Jüri Reimand [19,20], Geraldine Cancel-Tassin [21,22], Olivier Cussenot[21,22], Chris M. Hovens[23,24,25], Niall M. Cocoran[23,24,25], Phillip D. Stricker [26], Thorsten Schlomm [27], Gail S. Prins[28], Karina Dalsgaard Sørensen [29,30], Pan Prostate Cancer Group*, HEROIC PCaPH Africa1K*, Joachim Weischenfeldt [7,27], Shingai B. A. Mutambirwa[11], Peter M. Ngugi[13], David M. Thomas[31], Zsofia Kote-Jarai [5], Rosalind A. Eeles [5,32], M. S. Riana Bornman [2] & Vanessa M. Hayes [1,2,16,17] ✉

[1]Ancestry & Health Genomics Laboratory, Charles Perkins Centre, School of Medical Sciences, Faculty of Medicine and Health, University of Sydney, Camperdown, NSW, Australia. [2]School of Health Systems and Public Health, Faculty of Health Sciences, University of Pretoria, Pretoria, South Africa. [3]National Health Laboratory Services, Johannesburg, South Africa. [4]Department of Anatomical Pathology, School of Pathology, University of the

Witwatersrand, Johannesburg, South Africa. [5]The Institute of Cancer Research, London, UK. [6]Childrens Cancer Institute, Lowy Cancer Centre, University of New South Wales Sydney, Randwick, NSW, Australia. [7]Biotech Research & Innovation Centre & Finsen Laboratory, University of Copenhagen, Rigshospitalet, Copenhagen, Denmark. [8]Computational Genomics Group, Charles Perkins Centre, School of Medical Sciences, Faculty of Medicine and Health, University of Sydney, Camperdown, NSW, Australia. [9]Department of Physiology, Faculty of Health Sciences, University of Pretoria, Pretoria, South Africa. [10]Department of Urology, University of Pretoria, Pretoria, South Africa. [11]Department of Urology, Sefako Makgatho Health Science University, Dr George Mukhari Academic Hospital, Medunsa, Ga-Rankuwa, South Africa. [12]Tshilidzini Hospital, Shayandima, Thohoyandou, Limpopo, South Africa. [13]Department of Urology, East African Kidney Institute, University of Nairobi, Nairobi, Kenya. [14]Department of Human Pathology, University of Nairobi, Nairobi, Kenya. [15]Department of Pathology and Laboratory Medicine, Weil Cornell Medicine, New York, NY, USA. [16]Manchester Cancer Research Centre, University of Manchester, Manchester M20 4GJ, UK. [17]Norwich Medical School, University of East Anglia, Norwich, UK. [18]The Earlham Institute, Norwich Research Park, Norwich, UK. [19]Computational Biology Program, Ontario Institute for Cancer Research, Toronto, ON, Canada. [20]Department of Medical Biophysics & Department of Molecular Genetics, University of Toronto, Toronto, ON, Canada. [21]CeRePP, Hospital Tenon, Paris, France. [22]Sorbonne Universite, GRC n°5 Predictive Onco-Urology, APHP, Tenon Hospital, Paris, France. [23]Collaborative Center for Genomic Cancer Medicine University of Melbourne, The Victorian Comprehensive Cancer Centre, Parkville, VIC, Australia. [24]Department of urology, Royal Melbourne Hospital, Melbourne, Parkville, VIC, Australia. [25]Department of Surgery, The University of Melbourne, Parkville, VIC, Australia. [26]St Vincent's Prostate Cancer Research Centre, Sydney, NSW, Australia. [27]Charité Universitätsmedizin Berlin, Berlin, Germany. [28]Department of Urology, University of Illinois at Chicago, Chicago, IL 60612, USA. [29]Department of Molecular Medicine, Aarhus University Hospital, Aarhus N, Denmark. [30]Department of Clinical Medicine, Aarhus University, Aarhus N, Denmark. [31]Centre for Molecular Oncology, School of Biomedical Sciences, University of New South Wales Sydney, Randwick, NSW, Australia. [32]The Royal Marsden NHS Foundation Trust London, London, UK. *Lists of authors and their affiliations appear at the end of the paper. ✉e-mail: vanessa.hayes@sydney.edu.au

## Pan Prostate Cancer Group

Rosalind A. Eeles [5,32], Colin S. Cooper[5,17], G. Steven S. Bova[33], Daniel S. Brewer [17,18], Robert G. Bristow [16], Mark N. Brook[5], Benedict Brors[34,35], Daniel Burns[5], Adam Butler[36], Geraldine Cancel-Tassin [21,22], Kevin C. L. Cheng[19,37], Niall M. Corcoran[23,24,25], Olivier Cussenot[21,22], Francesco Favero[38,39], Clarissa Gerhauser[34], Abraham Gihawi[17], Etsehiwot G. Girma[38,39], Vincent J. Gnanapragasam[40], Andreas J. Gruber[41], Anis Hamid[25], Vanessa M. Hayes[1,2,16,17], Housheng Hansen He[42], Chris M. Hovens[23,24,25], Eddie Luidy Imada[15], G. Maria Jakobsdottir[16,43], Weerachai Jaratlersiri[1,8], Jue Jiang[1], Chol-Hee Jung[44], Francesca Khani[15], Daria Kiriy[7], Zsofia Kote-Jarai [5], Philippe Lamy[29,30], Gregory Leeman[34], Massimo Loda [15], Pavlo Lutsik[34,45], Luigi Marchionni[15], Ramyar Molania[46,47], Anthony T. Papenfuss[46,47], Diogo Pellegrina[19], Bernard Pope[44,48], Lucio R. Queiroz[15], Tobias Rausch[49], Jüri Reimand[19,37], Brain Robinson[15], Atef Sahli[16], Thorsten Schlomm [27], Pamela X. Y. Soh[1], Karina Dalsgaard Sørensen [29,30], Sebastian Uhrig[34], David C. Wedge [16], Joachim Weischenfeldt [7,27], Yaobo Xu[36], Takafumi N. Yamaguchi[50] & Claudio Zanettini[15]

[33]Prostate Cancer Research Center, Faculty of Medicine and Health Technology, Tampere University, Tampere, Finland. [34]German Cancer Research Center (DKFZ), Heidelberg, Germany. [35]Medical Faculty Heidelberg and Faculty of Biosciences, Heidelberg University, Heidelberg, Germany. [36]Wellcome Sanger Institute, Cambridge, UK. [37]Department of Medical Biophysics, University of Toronto, Toronto, Canada. [38]Biotech Research and Innovation Centre, University of Copenhagen, Copenhagen, Denmark. [39]Finsen Laboratory, Copenhagen University Hospital Rigshospitalet, Copenhagen, Denmark. [40]Department of Surgery, University of Cambridge and Cambridge University Hospitals NHS Trust, Cambridge Biomedical Campus, Addenbrooke's Hospital, Cambridge, UK. [41]University of Konstanz, Konstanz, Germany. [42]Princess Margaret Cancer Centre, University Health Network, Department of Medical Biophysics, University of Toronto, Toronto, Canada. [43]Christie Hospital, The Christie NHS Foundation Trust, Manchester Academic Health Science Centre, Manchester, UK. [44]Melbourne Bioinformatics, The University of Melbourne, Melbourne, Australia. [45]Department of Oncology, KU Leuven, Leuven, Belgium. [46]Walter and Eliza Hall Institute of Medical Research, Melbourne, Australia. [47]Department of Medical Biology, The University of Melbourne, Melbourne, Australia. [48]Australian BioCommons, The University of Melbourne, Melbourne, Australia. [49]Genome Biology, European Molecular Biology Laboratory (EMBL), Heidelberg, Germany. [50]Jonsson Comprehensive Cancer Center, University of California Los Angeles, Los Angeles, CA, USA.

## HEROIC PCaPH Africa1K

Vanessa M. Hayes[1,2,16,17], M. S. Riana Bornman[2], Peter Mungai Ngugi[13], Gail S. Prins[28], Weerachai Jaratlersiri[1,8], Winstar M. Ombuki[13], Sean M. Patrick[2], Daniel M. Moreira[28], Ikenna C. Madueke[28], Maria Argos[51], Irene E. J. Barnhoorn[52], Lynn Birch[28], Daniel S. Brewer [17,18], Robert G. Bristow [16], Raymond A. Campbell[10], Colin S. Cooper[5,17], Jenna Craddock[1,2], Rosalind A. Eeles [5,32], G. Nicolo' Fanelli[15], Eva Ferlev Jensby[1,29], Hagen E. A. Förtsch[53], Jessie Gamxamub[53], Kazzem Gheybi[1], Abraham Gihawi[1,17], Tingting Gong[1], Md. Mehedi Hasan[1], Vivien Holmes[16], Ruotian Huang[1], Jue Jiang[1], Zsofia Kote-Jarai[5], Maphuti Tebogo Lebelo[1,2,9], Massimo Loda [15], Melanie Louw[3,4], Pavlo Lutsik[45,54], Umuna Maendo[1,55], Tumisang M. N. Mbeki[2], Reginald Menoe[56], Shingai B. A. Mutambirwa[11], Muriuki Elias Nyaga[57], Micah O. Oyaro[14], Willis Oyieko[58], Joyce Shirinde[2], Pamela X. Y. Soh[1], Golda Stellmacher[53], Avraam Tapinos[16], Korawich Uthayopas[1], Douglas I. Walker[59], Edwin O. O. Walong[60], Githui Sheila Wanjiku[13], David C. Wedge [16], Allan Yienya[61] & Kangping Zhou[1]

[51]Department of Environmental Health, School of Public Health, Boston University, Boston, USA. [52]Department of Zoology, School of Natural and Mathematical Sciences, University of Venda, Thohoyandou, South Africa. [53]Windoek Central Academic Hospital, University of Namibia Medical Campus,

Windhoek, Namibia. [54]Division of Cancer Epigenomics, German Cancer Research Center (DKFZ), Heidelberg, Germany. [55]Botswana International University of Science and Technology, Palapye, Botswana. [56]Department of Urology, Rustenburg Hospital, Rustenburg, North West, South Africa. [57]Meru County Referral Hospital, Moi University, Meru, Meru County, Kenya. [58]Maseno University, Kisumu County, Kisumu, Kenya. [59]Gangarosa Department of Environmental Health, Rollins School of Public Health, Emory University, Atlanta, GA, USA. [60]Department of Pathology, University of Nairobi, Nairobi, Kenya. [61]Jaramogi Oginga Odinga Teaching and Referral Hospital, Kisumu, Kenya.

