## [Transparent Peer Review file · Nature Communications]

Pathogenic variants reveal candidate genes for prostate cancer germline testing in Black men

Corresponding Author: Professor Vanessa Hayes

Version 0:

Reviewer comments:

Reviewer #1

(Remarks to the Author)

The manuscript "Pathogenic variants reveal candidate genes for prostate cancer germline testing in Black men" presented a research that will fill critical gaps in our understanding of the specific germline variation that drives prostate cancer among African men. The study reported the spectrum of understudied germline pathogenic variants among indigenous African men who are most impacted by prostate cancer. Specifically, it identified prostate cancer risk alleles in genes like PREX2, POLE, FAT1, BRCA2, POLQ, LRP1B, ATM, and POLG, among others. In addition, several variants of unknown significance that are currently not captured in global genomic data repositories were reported. This is indeed a significant and original contribution to prostate cancer genomics, precision prostate cancer control and health equity research.

The data presented in the manuscript clearly supported the conclusion drawn by the authors. I, however, wonder if there are specific variants that are more enriched among specific ancestry, like West African vs East African genetic clusters. While it will be good for the authors to address this, I do think this publication is suitable for publication as it is. Finally, the robustness of the methodology employed by the authors is commendable.

Reviewer #2

(Remarks to the Author)

In a search for understanding aggressive prostate cancer in African/black patients, Gheybi et al have compiled a thorough analysis of germline variants using the Southern African Prostate Cancer Study, founding of HEROIC consortium, and the Pan Prostate Cancer Group. 217 index cases with 172 pathogenic variants identified within 78 DDR or prostate cancer related genes. However, the total of 11% significant variants is overall less than non-African populations, particularly genes such as BRCA1, HOXB13, CDK12, MLH1, MSH2, and BRIP1 seem unaffected in these patient populations. Variants in the DNA polymerases, as well as PREX2, POLE, FAT1, BRCA2, POLQ, LRP1B, and ATM, seem to be more prevalent in these patient populations.

Overall the data is well presented and the manuscript is clearly written. Some questions for the authors which, if answerable, may enhance the implications of this study to the prostate cancer field.

The authors note that SAPCS patients are more likely rural with lower health literacy and therefore may present with higher PSA levels and more advanced disease – is there any outcome data that these are linked to specific germline alterations more than the Pan-prostate cancer group? In addition, are the more advanced disease more associated with worse outcomes to standard androgen deprivation and second generation novel hormonal agents like enzalutamide or abiraterone? May be helpful to have these data.

The authors astutely point out the paradox of fewer germline variants found in this African population of known germline genes, while this patient population has a higher mortality – do the authors have a way of evaluating which of the variants of unknown significance may actually be pathogenic? In particular the DNA polymerase genes, and PREX2 gene seems found at higher rates and although not known pathogenic, could they be pathogenic in this patient population?

These are largely South Africa based patient populations. How applicable would these results be for African populations around the world?

For the most common gene found, a variant of PREX2, which the authors state is a DDR-relevant oncogene with a single splice donor genomic variant – how do the authors suggest further classifying this mutation?

The authors suggest whole genome sequencing as a more cost effective alternative to targeted sequencing – but would this be helpful or truly cost effective if mutations arise that are unknown whether they contribute to germline risk or other family risk?

For the suggested “African inclusive GT panel for prostate cancer”, what would be the suggested minimum set of genes to include for future studies? Particularly if trials are set up for these populations and a list of genes is needed for stratification?

Reviewer #3

(Remarks to the Author)

Reviewer #4

(Remarks to the Author)

Gheybi et al. collated WGS data from 217 men of African ancestry (n=146 published and 71 new cases) to perform a genome-wide interrogation of germline pathogenic variants. These data were compared against WGS data from 1) prostate cancer patients of non-African ancestry (959 men) and 2) from cancer-free individuals (92 Black men aged <50 years and 3209 White Australians aged >=75 years). The authors focused on single nucleotide variants (SNVs) and insertions and deletions (Indels). They reported 172 potentially pathogenic variants (PPV) in 78 genes related to prostate cancer or involved in DNA damage repair in 6% of the patients (13/217). This is well below the reported prevalence of germline PPVs in non-African prostate cancer patients (11-17%). When ranked according to variant frequency and functionality, clinical presentation and allelic status, PREX2, POLE, FAT1, BRCA2, POLQ, LRP1B and ATM were the top-ranking genes in their analysis. They also identify some DNA polymerases and DNA mismatch repair genes that were not as prevalent in previous reports. Although African ancestry is an established factor for incidence, higher disease stage and worse prognosis for prostate cancer, PPVs in Black men are understudied. As such, Black men do not benefit enough from germline testing panels that are created using genetic information from non-African populations. This is an important study which contributes to our understanding of the pathogenic germline variants in African prostate cancer patients and is likely to change the candidate genes to be included in germline testing.

The main weakness of this study is the lack of structural variants (SVs) and copy number aberrations (CNAs). Since these mutation types are not interrogated, the true prevalence of germline mutations can never be assessed. The reported prevalence of germline variants is still lower than that of non-African populations. I wonder if they could capture more variants if they included SVs and CNAs.

Given the availability of WGS data, the author should give a more thorough analysis of the genome-wide somatic mutations (TMB, MSI, mutational signatures in Indels, SNVs, SVs and CNAs) and their associations with germline variants considering their allelic status.

1. Do the tumors from individuals with germline mutations have mutational signatures known to be associated with the same genes (for example BRCA2)?
2. They also report some DNA polymerase variants. Were these associated with high TMB and particular mutational signatures in the somatic genome?
3. Were there any new mutational signatures observed in these tumors that might correlate with PPVs?

In step-3 of their analytical workflow, the authors try to identify genes that have not been reported to have pathogenic variants in ClinVar/InterVar. They use PolyPhen and SIFT to predict functional impact. These two tools perform poorly compared to more recent tools. The authors should try a tool like AlphaMissense [PMID: 37733863] that combines structure predictions from AlphaFold and sequence context. It would be interesting to see if the authors could capture more variants with this method.

Version 1:

Reviewer comments:

Reviewer #1

(Remarks to the Author)

The authors have responded satisfactorily to my comments.

Reviewer #2

(Remarks to the Author)

The authors have sufficiently addressed all reviewer comments. Unfortunately several requests for associations to outcomes or relevance across broader African populations are not available with the current dataset. Additions on associations with TMB and MSI status overall improve the relevance of the manuscript.

Reviewer #3

(Remarks to the Author)

Reviewer #4

(Remarks to the Author)

The authors responded to all the points I raise. I have no further questions.

RESPONSE TO REVIEWERS' COMMENTS

Responses: Blue

Changes to main text/supplement: red

Reviewer #1 (Remarks to the Author):

The manuscript “Pathogenic variants reveal candidate genes for prostate cancer germline testing in Black men” presented a research that will fill critical gaps in our understanding of the specific germline variation that drives prostate cancer among African men. The study reported the spectrum of understudied germline pathogenic variants among indigenous African men who are most impacted by prostate cancer. Specifically, it identified prostate cancer risk alleles in genes like PREX2, POLE, FAT1, BRCA2, POLQ, LRP1B, ATM, and POLG, among others. In addition, several variants of unknown significance that are currently not captured in global genomic data repositories were reported. This is indeed a significant and original contribution to prostate cancer genomics, precision prostate cancer control and health equity research.

Response: We thank the reviewer for their positive acknowledgement for the importance and relevance of this work.

The data presented in the manuscript clearly supported the conclusion drawn by the authors. I, however, wonder if there are specific variants that are more enriched among specific ancestry, like West African vs East African genetic clusters. While it will be good for the authors to address this, I do think this publication is suitable for publication as it is. Finally, the robustness of the methodology employed by the authors is commendable.

Response: As the study is biased towards men of Southern African ancestry (86.2%; 187/217), with the remaining 13.8% (30/217) reflecting a more mixed west African ancestral ancestry, searching for genetic clusters between African ancestries is currently not feasible. However, we suspect based on common variant data, as summarised by ourselves in a commentary (Soh PXY, Hayes VM. Eur Urol. 2023; 84(1):22-24), that there will likely be gene enrichment, while variant enrichment for rare variants may not be likely. It is notable that in this study we identified six low-frequency potentially pathogenic variants in our southern African patients, which were absent in our Southern and East African controls and rare in largely west African ancestral data (represented by the gnomAD west ancestral bias). These variants possibly reflect a southern African variant cluster requiring further investigation in African data across the diaspora. Currently such data is not available to the research community. We thank again the reviewer for acknowledging the robustness of our methodology and recommendation to publish.

Reviewer #2 (Remarks to the Author):

In a search for understanding aggressive prostate cancer in African/black patients, Gheybi et al have compiled a thorough analysis of germline variants using the Southern African Prostate Cancer Study, founding of HEROIC consortium, and the Pan Prostate Cancer Group. 217 index cases with 172 pathogenic variants identified within 78 DDR or prostate cancer related genes. However, the total of 11% significant variants is overall less than non-African populations, particularly genes such as BRCA1, HOXB13, CDK12, MLH1, MSH2, and BRIP1 seem unaffected in these patient populations. Variants in the DNA polymerases, as well as PREX2, POLE, FAT1, BRCA2, POLQ, LRP1B, and ATM, seem to be more prevalent in these patient populations.

Overall the data is well presented and the manuscript is clearly written. Some questions for the authors which, if answerable, may enhance the implications of this study to the prostate cancer field.

Response: We appreciate the reviewer's compliments regarding the presentation and writing of the paper and its relevance for the field.

The authors note that SAPCS patients are more likely rural with lower health literacy and therefore may present with higher PSA levels and more advanced disease – is there any outcome data that these are linked to specific germline alterations more than the Pan-prostate cancer group? In addition, are the more advanced disease more associated with worse outcomes to standard androgen deprivation and second generation novel hormonal agents like enzalutamide or abiraterone? May be helpful to have these data.

Response: The SAPCS cohort has little to no access to outcomes data, hence we are not able to address the relevance of pathogenic variants to outcomes as a variable. Secondly, as SAPCS patients present late, many do not undergo treatment, which is perpetuated by the mode of ADT applied being orchidectomy (reducing compliance) rather than hormonal agents as practiced outside the continent. In summary, this data is unfortunately not available.

The authors astutely point out the paradox of fewer germline variants found in this African population of known germline genes, while this patient population has a higher mortality – do the authors have a way of evaluating which of the variants of unknown significance may actually be pathogenic? In particular the DNA polymerase genes, and PREX2 gene seems found at higher rates and although not known pathogenic, could they be pathogenic in this patient population?

Response: They certainly could be pathogenic and would require further functional studies using models to predict pathogenicity and as such outside the scope of this study.

These are largely South Africa based patient populations. How applicable would these results be for African populations around the world?

Response: This is yet to be determined and forms the basis for further studies by the team to include other regions and ancestries across Sub-Saharan Africa and beyond. There is a severe shortage of whole genome data representing African populations globally.

For the most common gene found, a variant of PREX2, which the authors state is a DDR-relevant oncogene with a single splice donor genomic variant – how do the authors suggest further classifying this mutation?

Response: PREX2 presented as the most impacted gene, but not the single most predicted variant. In this study we developed a ranking system which allowed us to combine knowledge pertaining to the variant, patient clinical presentation as well as features of the tumour in a robust prediction of functionality. Any further assessments would call for the development of variant-specific models to predict functionality, each with their own limitations and extensive associated cost.

The authors suggest whole genome sequencing as a more cost effective alternative to targeted sequencing – but would this be helpful or truly cost effective if mutations arise that are unknown whether they contribute to germline risk or other family risk?

Response: As the cost of sequencing comes down, it is true that considering whole genome sequencing for germline testing requires consideration. This is further perpetuated by the significant heterogeneity for potential germline target genes identified in our African cohort. However, the reviewer is correct in that determining the actual risk for variants of unknown significance (VUS) including genes of unknown significance remains problematic. This study is therefore important, as it further highlights the importance for African data and how the lack of African data perpetuates African exclusion using current germline testing criteria.

For the suggested “African inclusive GT panel for prostate cancer”, what would be the suggested

minimum set of genes to include for future studies? Particularly if trials are set up for these populations and a list of genes is needed for stratification?

Response: As a 'first-of-its-kind' study we are a little way from developing an African inclusive GT panel. While this study provides a list of potential target genes, we call for further 'whole genome' African studies to build on this study. Ultimately, a larger number of African genomic data is required to develop a GT panel – this is an important first step.

Reviewer #3 (Remarks to the Author):

Response: Thank you for your contribution and for the primary Reviewer to embrace this important initiative which includes ECR's in the review process.

Reviewer #4 (Remarks to the Author):

Gheybi et al. collated WGS data from 217 men of African ancestry (n=146 published and 71 new cases) to perform a genome-wide interrogation of germline pathogenic variants. These data were compared against WGS data from 1) prostate cancer patients of non-African ancestry (959 men) and 2) from cancer-free individuals (92 Black men aged <50 years and 3209 White Australians aged >=75 years). The authors focused on single nucleotide variants (SNVs) and insertions and deletions (Indels). They reported 172 potentially pathogenic variants (PPV) in 78 genes related to prostate cancer or involved in DNA damage repair in 6% of the patients (13/217). This is well below the reported prevalence of germline PPVs in non-African prostate cancer patients (11-17%). When ranked according to variant frequency and functionality, clinical presentation and allelic status, PREX2, POLE, FAT1, BRCA2, POLQ, LRP1B and ATM were the top-ranking genes in their analysis. They also identify some DNA polymerases and DNA mismatch repair genes that were not as prevalent in previous reports. Although African ancestry is an established factor for incidence, higher disease stage and worse prognosis for prostate cancer, PPVs in Black men are understudied. As such, Black men do not benefit enough from germline testing panels that are created using genetic information from non-African populations. This is an important study which contributes to our understanding of the pathogenic germline variants in African prostate cancer patients and is likely to change the candidate genes to be included in germline testing.

Response: We thank the reviewer for acknowledging the importance of this study and further emphasising the importance of African inclusion in developing PCa specific germline testing panels and criteria.

The main weakness of this study is the lack of structural variants (SVs) and copy number aberrations (CNAs). Since these mutation types are not interrogated, the true prevalence of germline mutations can never be assessed. The reported prevalence of germline variants is still lower than that of non-African populations. I wonder if they could capture more variants if they included SVs and CNAs. Given the availability of WGS data, the author should give a more thorough analysis of the genome-wide somatic mutations (TMB, MSI, mutational signatures in Indels, SNVs, SVs and CNAs) and their associations with germline variants considering their allelic status.

Response: Currently germline testing panel studies of non-African data exclude for SV and CNV interrogation, this is largely due to the need for whole genome data for thorough interrogation, current limitations in accurate/consensus SV and CNV calling and lastly interpretation of SV/CNV germline data. We recently addressed this limitation for PCa germline testing for both African and non-African patients, at least for SVs in a recent publication in Nature Communications (Gong T, et al.

Rare pathogenic structural variants show potential to enhance prostate cancer germline testing for African men. **Nat Commun.** 2025 Mar 10;16(1):2400). Notably, the latter study overlaps with 1135 Southern African patients from this study, while in a next phase we are addressing CNVs. As current germline testing assays are limited to SNV and indel data, we feel this study appropriately focused on these variant types. The team is planning, however, to address germline CNVs in an independent paper, largely due to preliminary (unpublished data) showing limitations and complexity with merging data and interpretation of multi-CNV calling tools.

1. Do the tumors from individuals with germline mutations have mutational signatures known to be associated with the same genes (for example BRCA2)?
2. They also report some DNA polymerase variants. Were these associated with high TMB and particular mutational signatures in the somatic genome?

Response: As tumour-matched data is available for the 115 SAPCS patients derived from a single analytical analysis (Jaratlerdsiri W, et al. African-specific molecular taxonomy of prostate cancer. **Nature.** 2022 Sep;609(7927):552-559), based on the reviewers most insightful comments, we have elected to address TMB and MSI as it pertains to patients presenting with POL gene (DNA polymerase) PPV/POVs, while we further address for gene-relevant and DDR-like tumour-matched mutational signatures, which also cover points 1 and 2 above. Consequently, we have added the following section to the Results (in red in main text), which includes reference to two additional supplementary Tables S14 and S15, as well as two minimal statements within the Discussion.

The Results text reads as follows:

Southern African patient-matched tumour mutational burden and signatures

Besides tumour features linked directly to PPV/POV ranking, having observed an overall higher tumour mutational burden (TMB, 1.197 vs 1.061 mutations/Mb, Log10-transformed $t=2.5207$, $P=0.01308$) and enrichment of mutational signatures of unknown significance (10 vs 1) in our SAPCS versus European-derived tumours¹⁸, we further sought to correlate biologically relevant PPV/POV status with patient-matched TMB, with a focus on the PPV/POVs impacting the DNA polymerases, and tumour enrichment for signatures known to be associated with the same or similar largely DDR-related aetiologies. Ranking TMBs for all 115 SAPCS patients, 10/20 (50%) of DNA polymerase presenting PPV/POV patients presented with a TMB above the median (1.23 mutations/Mb) ranging from 1.53 to 3.31 mutations/Mb and including a single outlier UP2113 (59.61) with associated microsatellite instability (MSI) (**Table S14**). Of the three patients presenting with two POL gene PPV/POVs each, this included the TMB outlier patient with *POLE* (p.Pro99Leu) and *POLQ* (p.Leu232Ile), while KAL0074 with an above median TMB (1.598) presented with *POLE* (p.Ser864Cys) and *POLG* (p.Arg993Cys). Additionally, mutational signatures associated with TMB or DNA polymerase variants, such as single-base-substitution (SBS)9, SBS10 (all), SBS14 and double-base-substitution (DBS)3, were absent in our study.

While the *BRCA2* associated signature SBS3 was found to be enriched in a single SAPCS patient with no DDR/know PCa-associated PPV/POV germline variant, no enrichment was observed for signatures with associated DDR-related aetiology, including SBS6, SBS15, SBS21, SBS26 and SBS44, while the *MSH6* POV carrier did not present with the gene-associated copy-number (CN)25 tumour enrichment. Conversely, 22 PPV/POV-presenting SAPCS patients harboured DDR-like mutational signatures (**Table S15**), including DBS7 (defective DNA mismatch repair), insertion-deletion (ID)1 and ID2 (defective DNA mismatch repair/DNA replication slippage), ID6 (homologous recombination DNA damage repair

associated with *BRCA2/1* mutations), ID8 (repair of DNA double strand breaks by non-homologous DNA end-joining mechanisms) and structural-variation (SV)³ (homologous recombination deficiency), of which 9/22 (40.9%) or 9/20 (45%, excluding for MAF/VAF criteria) presented with two or more PPV/POVs. Notably, two patients with POLQ POVs (p.Arg784Cys and p.Ser1618X) showed enrichment for both ID1 and SV3.

Supplementary Tables

Table S14. SAPCS patients presenting with DNA polymerase PPV/POVs (n=20) ranked by patient-matched tumour mutational burden (TMB, highest to lowest) and including evidence for microsatellite instability (MSI).

POL Gene	AA Change	PPV/POV	Patient ID^a	TMB	MSS/MSI
POLE	P99L	POV	UP2113^{R1}	59.61010363	MSI-H
POLQ	L232I	PSLF-POV	UP2113^{R2}	59.61010363	MSI-H
POLE	R1297C	POV	KAL070	3.31444300	MSS
POLE	P99L	POV	UP2050 ^{R1}	3.00356217	MSS
POLG	R562G	POV	UP2039	2.56088082	MSS
POLQ	L232I	PSLF-POV	UP2116 ^{R2}	2.10200777	MSS
POLE	E1554K	POV	N0067	2.05246114	MSS
POLQ	R784C	POV	SMU030 ^{R3}	1.87726683	MSS
POLQ	R784C	POV	N0053 ^{R3}	1.84455958	MSS
POLE	S864C	POV	KAL0074	1.59844559	MSS
POLG	R993C	POV	KAL0074	1.59844559	MSS
POLQ	S1618X	POV	SMU050	1.53076424	MSS
POLE	P99L	POV	UP2197 ^{R1}	1.14831606	MSS
POLQ	L232I	PSLF-POV	UP2004^{R2}	1.02040155	MSS
POLQ	L388del	POV	UP2004	1.02040155	MSS
POLQ	K83fs	POV	KAL0106	1.00680051	MSS
POLQ	L232I	PSLF-POV	SMU097 ^{R2}	0.76878238	MSS
POLG	F749S	PSLF-PPV	SMU177 ^{R4}	0.46599741	MSS
POLG	G531S	POV	UP2317	0.37240932	MSS
POLG	F749S	PSLF-PPV	KAL0101 ^{R4}	0.07059585	MSS
POLG	F749S	PSLF-PPV	SMU159 ^{R4}	0.0625	MSS
POLE	R1355C	POV	SMU111	0.0625	MSS
POLQ	L232I	PSLF-POV	UP2092 ^{R2}	0.03465025	MSS

^aPatient IDs in bold presenting with more than one POL gene PPV/POV (3 patients). ^{R1-R4}Represent patients (by ID) presenting with the same POL gene pathogenic variant.

Abbreviations: AA, amino acid; PPV, potentially pathogenic variant; POV, potentially oncogenic variant; PSLF, population specific low-frequency; TMB, tumour mutational burden; MSS, microsatellite stability; MSI-H, microsatellite instability high

Table S15. SAPCS patients (n=22) presenting with potentially pathogenic germline variants (PPV/POVs) and showing tumour-matched enrichment for DNA damage repair (DDR)-like mutational signatures.

Gene	AA change	PPV/POV	Ranking	Patient Tumour ID	Mutational Signature
BRCA2	W31R	PPV	4	UP2103	ID6
PREX2	R1230W	PSLF-POV	8		
FANCA	R504G	POV	4.5	N0084	ID1
FANCG	YRQ213-215X	PPV	5.5	UP2093	DSB7

PREX2	R1230W	PSLF-POV	8		
RECQL4	AGR805-807del	PPV	4	UP2187	DSB7
CLSPN	Splice donor	POV	3.5		
ERCC4	I706T	POV	2.5	N0078	ID1
RAD23B	QG338-339R	POV	2.5		
POLE	P99L	POV	7.5	UP2113	ID2
POLQ	L232I	LF-POV	19		
FAT1	A1865V	POV	0.5	SMU030	ID1 & SV3
POLQ	R784C	POV	2.5		
HERC2	A332S	POV	1.5	KAL0091	SV3
LEF1	R89W	POV	1.5		
PREX2	R1230W	PSLF-POV	8		
RET	V871I	POV	2.5	SMU041	SV3
KMT2A	R3242Q	POV	3.5		
MUTYH	splice donor	POV	3	SMU097	SV3
POLQ	L232I	PSLF-POV	19		
FANCI	splice donor	POV	4	UP2330	ID2
TRRAP	A2335G	POV	8	UP2133	ID1
RTEL1	R898C	POV	7	UP2258	ID1
LRPIB	Y3183C	POV	2	UP2396	ID6
ERBB3	R838Q	PPV	3	N0007	ID8
POLQ	S1618X	POV	5.5	SMU050	ID1 & SV3
FAT1	G331D	POV	2	UP2372	SV3
ERCC4	A860D	POV	MAF excluded	SMU167	SV3
KMT2A	K3181R	POV	2	KAL0078	SV3
PREX2	K787E	PSLF-POV	9	N0001	SV3
ERCC2	R608C	POV	2.5	N0088	SV3
DNMT3A	R326P	POV	VAF excluded	SMU083	SV3

Abbreviations: AA, amino acid; PPV, potentially pathogenic variant; POV, potentially oncogenic variant; PSLF, population specific low-frequency; ID, insertion-deletion; DBS, double-base-substitution; SV, structural-variation; CN, copy-number.

Additions to the Discussion

From line 413: Additionally, we found 50% of the tumour-matched SAPCS *POLE*, *POLQ*, and *POLG* carriers to present with an above median TMB.

From line 451: Additionally, while candidate PPV/POVs in known GT genes, including *BRCA2* (p.Trp31Arg) and *FANCA* (p.Arg504Gly) showed tumour associated DDR-like mutational signature enrichment, the 28 PPV/POVs in unknown GT-candidates showing DDR-like mutational enrichment require further consideration.

3. Were there any new mutational signatures observed in these tumors that might correlate with PPVs?

Response: Having reported 10 new mutational signatures significantly enriched in our African vs European tumours (Jaratlerdsiri et al., Nature 2022), as these are currently of unknown aetiology, speculating further regarding PPV/POV association, we believe would 'currently with minimal African-specific data' add no additional value. However, this will be addressed with the inclusion of further African-inclusive data and we appreciate the suggestion.

In step-3 of their analytical workflow, the authors try to identify genes that have not been reported to

have pathogenic variants in ClinVar/InterVar. They use PolyPhen and SIFT to predict functional impact. These two tools perform poorly compared to more recent tools. The authors should try a tool like AlphaMissense [PMID: 37733863] that combines structure predictions from AlphaFold and sequence context. It would be interesting to see if the authors could capture more variants with this method.

Response: This is certainly true, As such a prior to this data analysis, this team in a recent publication by Zhou K, Gheybi K, Soh PXY, Hayes VM. Evaluating variant pathogenicity prediction tools to establish African inclusive guidelines for germline genetic testing. **Commun Med (Lond)**. 2025 May 6;5(1):157 (see Supplementary Figure as below), addressed this very concept, testing all 54 variant pathogenic prediction tools (VPPT), showing PolyPhen2 and SIFT to outperform AlphaMissense for our African data (orange). In fact, AlphaMissense ranked 50/54 for the VPPT tools for African-specific data and of relevance to this study, especially Southern African biased data. Based on our findings, we retain our use of PolyPhen2 and SIFT.

RESPONSE TO REVIEWERS' COMMENTS

Reviewer #1 (Remarks to the Author):

The authors have responded satisfactorily to my comments.

Response: appreciated

Reviewer #2 (Remarks to the Author):

The authors have sufficiently addressed all reviewer comments. Unfortunately several requests for associations to outcomes or relevance across broader African populations are not available with the current dataset. Additions on associations with TMB and MSI status overall improve the relevance of the manuscript.

Response: appreciated

Reviewer #3 (Remarks to the Author):

Response: appreciated and hope it was a good experience

Reviewer #4 (Remarks to the Author):

The authors responded to all the points I raise. I have no further questions.

Response: appreciated